# THAT ESCALATED QUICKLY: COMPOUNDING COMPLEXITY BY EDITING LEVELS AT THE FRONTIER OF AGENT CAPABILITIES

## ABSTRACT

Deep Reinforcement Learning (RL) has recently produced impressive results in a series of settings such as games and robotics. However, a key challenge that limits the utility of RL agents for real-world problems is the agent's ability to generalize to unseen variations (or *levels*). To train more robust agents, the field of *Unsupervised Environment Design* (UED) seeks to produce a curriculum by updating both the agent and the distribution over training environments. Recent advances in UED have come from promoting levels with high *regret*, which provides theoretical guarantees in equilibrium and empirically has been shown to produce agents capable of zero-shot transfer to unseen human-designed environments. However, current methods require either learning an environment-generating adversary, which remains a challenging optimization problem, or curating a curriculum from randomly sampled levels, which is ineffective if the search space is too large. In this paper we instead propose to *evolve* a curriculum, by making edits to previously selected levels. Our approach, which we call *Adversarially Compounding Complexity by Editing Levels* (ACCEL), produces levels at the frontier of an agent's capabilities, resulting in curricula that start simple but become increasingly complex. ACCEL maintains the theoretical benefits of prior works, while outperforming them empirically when transferring to complex out-of-distribution environments.

## 1 INTRODUCTION

Reinforcement Learning (RL, Sutton & Barto (1998)) considers the problem of an agent learning from experience in an environment to maximize total (discounted) of reward. The past decade has seen a surge of interest in RL, with high profile successes in games (Vinyals et al., 2019; Berner et al., 2019; Silver et al., 2016; Mnih et al., 2013; Hu & Foerster, 2020) and robotics (OpenAI et al., 2019; Andrychowicz et al., 2020). As such, there is tremendous excitement that RL may be a path towards generally capable agents (Silver et al., 2021). Despite these successes, deploying RL agents in the real world remains a challenge (Dulac-Arnold et al., 2019). Notably, strong training performance in simulation may not result in policies that are robust to the many sources of variation in the real world.

Addressing this challenge on the agent side has become an active area of research (Zhang et al., 2021a; Agarwal et al., 2021a; Raileanu & Fergus, 2021), but in this paper we instead focus on the impact of the *training environment* itself, which often has a significant impact on agent's ability to generalize (Co-Reyes et al., 2020). For example in locomotion tasks, Reda et al. (2020) found that the initial state distribution, survival bonus, reward structure and control frequency had a significant impact on the performance of an agent. Indeed, *curricula* over environments can also influence the generalization performance of the agent (Jiang et al., 2021b). Throughout this paper we consider distributions of environments, referring to each individual sample as a *level*. Given a parameterized environment, the simplest approach one can consider is Domain Randomization (DR, Jakobi, 1997; Tobin et al., 2017; Sadeghi & Levine, 2017; Risi & Togelius, 2020; Peng et al., 2017), whereby an agent trains on individual levels uniformly sampled from an underlying environment distribution. It has been shown that training an agent with a DR-type approach can produce agents capable of complex real-world skills (OpenAI et al., 2019). However, the performance of DR is only as good as the sampling distribution available—thus it can be ineffective when the probability of sampling useful levels is too low.

Figure 1: The evolution of a level: At first, the editor places blocks outside the trajectory of the optimal policy, which acts as an augmentation as the agent has a fully observable view. Then, the editor moves the agent further from the goal, before placing challenging obstacles in its path. Note that since the agent can move diagonally in this environment, the final level is solvable. Each level is a high Positive Value Loss at the time it is included in the level store, thus the level co-evolves with the agent over time.

Recently, *Unsupervised Environment Design* (UED, Dennis et al., 2020) has emerged as formalism for methods to design effective curricula. Given a parameterized environment, UED methods frame learning as a game between a *teacher* which generates a curriculum of levels, and a *student* seeking to maximize some notion of return. UED is a generalization of several other approaches. Indeed, DR can be considered as a UED algorithm whereby the teacher generates environments uniformly at random from the environment distribution. Other approaches to UED consider learning a teacher agent (or generator), with a variety of adversarial objectives proposed (Dennis et al., 2020; Gur et al., 2021). However, training a teacher is a challenging optimization problem, suffering from both nonstationarity and sparse reward, as the teacher's feedback only comes after evaluation by a changing student policy. Recent work showed it can be more effective to simply *curate* levels produced by DR (Jiang et al., 2021b;a; Matiisen et al., 2020), producing a curriculum of increasingly complex randomly generated levels. Despite their promise, these methods can only be as effective as the best of the random levels they sample, which can be a limitation in high dimensional design spaces. Finally, another series of promising works seek to *evolve* populations of environments (Wang et al., 2019; 2020; Dharna et al., 2020), but these methods heavily rely on handcrafted heuristics and also use up to 20x more compute since they also train a population of agents. In this paper, we seek a general method which harnesses the benefits of all three of these approaches. We posit the following: *Rather than generate levels from scratch, it may be more effective to edit previously curated levels*.

Our primary contribution is to propose a new method which we call *Adversarially Compounding Complexity by Editing Levels*, or ACCEL. ACCEL is an evolutionary process, with levels constantly changing to remain at the frontier of the student agent's capabilities (see: Figure 2).

As such, levels generated by ACCEL begin simple but quickly become more complex. This benefits both the beginning of training (Berthouze & Lungarella, 2004), as the student begins learning much faster, while it also facilitates the construction of complex structures (see Figure 1). We believe ACCEL provides the best of both worlds: an evolutionary approach that can generate increasingly complex environments, combined with a regret-based curator which provides theoretical robustness guarantees in equilibrium. We evaluate ACCEL on a series of challenging procedurally generated grid world environments, where ACCEL demonstrates the ability to rapidly increase complexity while maintaining performance.

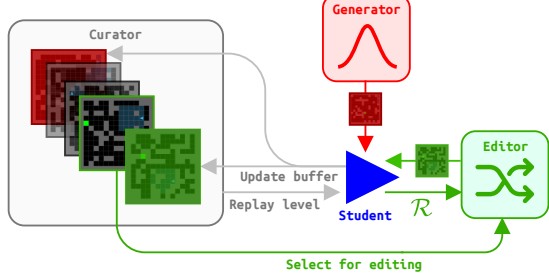

Figure 2: An overview of ACCEL. Levels are (randomly) sampled from a generator, and evaluated, with high regret levels added to the level buffer. The curator selects levels to replay, which are used to train the student agent. After training, the levels are passed to the editor and the edited levels are added to the level store if they are high regret.

Finally, we show ACCEL makes it possible to train agents capable of transfer to mazes an order of magnitude larger than training levels, achieving over double the success rate of the next best baseline.

## 2 BACKGROUND

### 2.1 FROM MDPs TO UNDERSPECIFIED POMDPs

A Markov Decision Process (MDP) is defined as a tuple $\langle S, A, \mathcal{T}, \mathcal{R}, \gamma \rangle$ where $S$ and $A$ stand for the sets of states and actions respectively and $\mathcal{T} : S \times A \to \mathbf{\Delta}(S)$ is a transition function representing the probability that the system/agent transitions from a state $s_t \in S$ to $s_{t+1} \in S$ given action $a_t \in A$. Each transition also induces an associated reward $r_t$ generated by a reward function $\mathcal{R} : S \to \mathbb{R}$, and

$\gamma$ is a discount factor. When provided with an MDP, the goal of Reinforcement Learning (RL, Sutton & Barto, 1998) is to learn a policy $\pi$ that maximizes expected discounted reward, i.e. $\mathbb{E}[\sum_{i=0}^{T} r_t \gamma^t]$.

Despite the generality of the MDP framework, it is often an unrealistic model for real world environments. First, it assumes full observability of the state, which is often impossible in practice. This is addressed in *partially observable* MDPs, or POMDPs, which include an observation function $\mathcal{I} : S \rightarrow O$ which maps the true state (which is unknown to the agent) to a (potentially noisy) set of observations $O$. Secondly, the traditional MDP framework assumes a single reward and transition function, which are fixed throughout learning. Instead, in the real world, agents may experience variations not seen during training, which makes it crucial that policies are capable of robust transfer.

To address both of these issues, we use the recently introduced *Underspecified* POMDP, or UPOMDP, given by $\mathcal{M} = \langle A, O, \Theta, S^{\mathcal{M}}, \mathcal{T}^{\mathcal{M}}, \mathcal{I}^{\mathcal{M}}, \mathcal{R}^{\mathcal{M}}, \gamma \rangle$. This definition is identical to a POMDP with the addition of $\Theta$ to represent the free parameters of the environment, similar to the context in a Contextual MDP (Modi et al., 2017). These parameters can be distinct at every time step and incorporated into the transition function $\mathcal{T}^{\mathcal{M}} : S \times A \times \Theta \rightarrow \mathbf{\Delta}(S)$. Following Jiang et al. (2021a) we define a *level* $\mathcal{M}_\theta$ as an environment resulting from a fixed $\theta$. We define the value of $\pi$ in $\mathcal{M}_\theta$ to be $V^\theta(\pi) = \mathbb{E}[\sum_{i=0}^{T} r_t \gamma^t]$ where $r_t$ are the rewards achieved by $\pi$ in $\mathcal{M}_\theta$. UPOMDPs benefit from their generality, since $\Theta$ can represent possible dynamics (for example in sim2real (Peng et al., 2017; OpenAI et al., 2019; Andrychowicz et al., 2020)), changes in observations, different reward functions or differing game maps in procedurally generated environments.

## 2.2 METHODS FOR UNSUPERVISED ENVIRONMENT DESIGN

The goal of Unsupervised Environment Design (UED, Dennis et al., 2020) is to generate a series of levels that form a curriculum for a student agent, such that the student agent is capable of transfer, by maximizing some utility function $U_t(\pi, \theta)$. In the case of DR, the utility function is simply:

$$U_t^U(\pi, \theta) = C \tag{1}$$

for any constant $C$. When learning a teacher, recent approaches proposed to use objectives seeking to maximize *regret*, defined as the difference between the expected return of the current policy and the optimal policy, ie:

$$U_t^R(\pi, \theta) = \underset{\pi^* \in \Pi}{\mathrm{argmax}}\{\mathrm{REGRET}^\theta(\pi, \pi^*)\} = \underset{\pi^* \in \Pi}{\mathrm{argmax}}\{V^\theta(\pi^*) - V^\theta(\pi)\} \tag{2}$$

Unlike other objectives, which may promote unsolvable environments, regret-based objectives have been shown to promote the simplest possible environments that the agent cannot currently solve (Dennis et al., 2020) in a range of settings. However, since we do not have access to $\pi^*$, a key challenge in UED algorithms utilizing objectives inspired by Equation 2 is to *approximate* the regret. Recently, the *Prioritized Level Replay* (PLR, Jiang et al., 2021b;a) algorithm introduced an additional teacher agent in the form of a *curator*, forming a "dual curriculum game". The curator maintains a buffer of previously experienced levels and selects levels to be replayed by the student policy using objectives approximating regret. One of the objectives used by PLR is *Positive Value Loss*, given by:

$$\frac{1}{T} \sum_{t=0}^{T} \max \left( \sum_{k=t}^{T} (\gamma\lambda)^{k-t} \delta_k, 0 \right) \tag{3}$$

where $\lambda$ and $\gamma$ are the Generalized Advantage Estimation (GAE, Schulman et al. (2016)) and MDP discount factors respectively, and $\delta_t$, the TD-error at timestep $t$. Since Positive Value Loss approximates regret, if the student *trains solely on curated levels* (i.e. does not take gradient steps on levels from the generator), then PLR achieves robustness guarantees in equilibrium. More formally, if $S_t = \Pi$ is the strategy set of the student and $S_t = \Theta$ is the strategy set of the teacher (in this case the curator), then (by Corollary 1 of Jiang et al. (2021a)), in equilibrium the resulting student policy $\pi$ converges to a minimax regret policy, ie:

$$\pi = \underset{\pi_A \in \Pi}{\mathrm{argmin}}\{ \underset{\theta, \pi_B \in \Theta, \Pi}{\max}\{\mathrm{REGRET}^\theta(\pi_A, \pi_B)\}\} \tag{4}$$

Empirically PLR has also been shown produce policies with strong generalization capabilities[1], yet it's main weakness is that it still relies on randomly sampling useful levels. Next, we introduce our new approach which seeks to leverage the curator to produce batches of high regret levels.

---

[1]To see the impact of PLR on a simple example, we include a visualization in Figure 19 in the Appendix.

## 3 COMPOUNDING COMPLEXITY BY EDITING LEVELS

In this section we introduce our new method for UED, building on regret-based methods such as PLR. As the dimensionality of the design space increases, it becomes increasingly challenging to randomly sample effective levels for learning—a problem we call "the curse of dimensionality in UED". Thus, rather than solely rely on curating random levels, we instead look to evolution to produce new batches of levels, by making edits to previously curated ones. This is a direct attempt to produce more levels at the "frontier" of agent capabilities, which has been shown to be useful in a variety of recent works (Wang et al., 2019; Jiang et al., 2021b; Zhang et al., 2020). Evolutionary methods are suitable for this, yet often require heuristics to define a workable fitness function. For example, POET pre-filters levels using handcrafted criteria to have a reward in the range $[50, 300]$. We propose a more general approach using regret in the form of Positive Value Loss to assess learning potential. We call our method *Adversarially Compounding Complexity by Editing Levels*, or ACCEL.

The key idea of ACCEL is to introduce an *editor*, which produces new levels for the agent by making edits to levels previously sampled by the curator. Editing involves making a handful of changes (e.g. adding/removing tiles on a maze), but could be extended to generative models (e.g. perturbations in a latent space). Equipped with the ability to edit levels, it is possible to produce an entire batch of useful levels from a single example, while incrementally increasing complexity. We consider both a learned editor, optimizing for Positive Value Loss (Equation 3), and a random one. Following Robust PLR (Jiang et al., 2021a) we do not initially train on edited levels. Instead, we evaluate them and only add them to the replay buffer if they meet the threshold for the scoring function (high regret). We consider two different criteria for selecting which replay levels to edit: those which the agent can now solve with low future learning potential, approximated as return minus regret, which we call "easy", and "batch" where we use the entire batch. The full procedure is shown in Algorithm 1.

---

**Algorithm 1** ACCEL (changes w.r.t Robust PLR)

**Input:** Buffer size $K$, initial fill ratio $\rho$, level generator.
**Initialize:** Initialize policy $\pi(\phi)$, level buffer $\Lambda$.
*# Initial Data Collection*
Sample $K * \rho$ initial levels.
*# Main Training Loop*
**while** *not converged* **do**
    Sample replay decision $d \sim P_D(d)$
    **if** $d = 0$ **then**
        Sample level $\theta$ from level generator
        Collect $\pi$'s trajectory $\tau$ on $\theta$, with a stop-gradient $\phi_\perp$
        Compute PLR score, $S = \mathbf{score}(\tau, \pi)$
        Add $\theta$ to $\Lambda$ if score $S$ meets threshold
    **else**
        Sample a replay level, $\theta \sim \Lambda$
        Collect policy trajectory $\tau$ on $\theta$
        Update $\pi$ with rewards $\boldsymbol{R}(\tau)$
        Edit $\theta$ to produce $\theta'$
        Collect $\pi$'s trajectory $\tau$ on $\theta'$, with a stop-gradient $\phi_\perp$
        Compute PLR score, $S = \mathbf{score}(\tau, \pi)$
        Add $\theta'$ to $\Lambda$ if score $S$ meets threshold
        (Optionally) Update Editor using score $S$
    **end**
**end**

---

We posit that editing is effective for two reasons. First, small incremental changes to a level can lead to a diverse batch of new ones (Sturtevant et al., 2020), which may move those that are currently too hard or too easy towards the frontier of the agent's capabilities. This may also prevent overfitting, for example, in Figure 3 we see three levels generated by ACCEL in a grid world environment (Chevalier-Boisvert et al., 2018). Each is an edit of the same level, and has a similar initial obser-

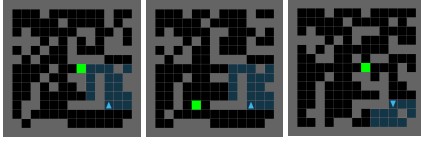

Figure 3: Levels generated by ACCEL. Though all levels are evolved from the same DR level, they require different behaviors to solve. Left: the agent can go up or left and reach the goal. Middle: the goal is on the left, while on the right the left path is blocked.

vation, yet requires the agent to explore in a different fashion to reach the goal. Training on these environments simultaneously will teach the agent to actively explore the environment. Second, making edits *outside* of the direct trajectory of the agent can be seen as a form of data augmentation, which has been shown to improve sample efficiency in RL (Laskin et al., 2020; Kostrikov et al., 2021; Raileanu et al., 2020), since it changes the observation but not the optimal policy.

ACCEL is an Evolutionary algorithm, whereby the "fitness" is (approximate) regret, since levels only stay in the "population" (or level replay buffer) if they meet the criteria for curation. Evolution has led to many successes in other domains (Stanley et al., 2019; Pugh et al., 2016), and even proven useful for UED with the POET algorithm. Compared to POET we have two key differences: first, we have a population of levels but not a population of agents, thus we have a single, generally capable

agent. This likely leads to a lower computational cost. In addition, ACCEL uses a minimax *regret* (rather than minimax) objective, which means we do not need handcrafted rules to select levels, since optimizing for regret naturally promotes levels at the frontier of agent's capability. Indeed, training on high regret levels also means that ACCEL inherits the robustness guarantees in equilbrium from Robust PLR (Corrollary 1 in Jiang et al. (2021a)):

**Remark 1.** *If the procedure described in Algorithm 1 finds a Nash equilibrium, then the student policy is following a minimax regret strategy.*

This is in stark contrast with other evolutionary approaches, which rely solely on empirical results.

## 4 EXPERIMENTS

In our experiments we seek to answer the following two questions: 1) Can ACCEL lead to sample efficient learning in complex design spaces? 2) Can ACCEL compound complexity to asymptotically produce agents capable of zero-shot transfer to challenging out-of-distribution environments? We conduct a series of experiments in grid world environments, as have been used in previous UED works (Dennis et al., 2020; Jiang et al., 2021a). These environments are made challenging by high dimensional observations and sparse rewards, thus they are often used to test state-of-the-art exploration methods (Raileanu & Rocktäschel, 2020; Zhang et al., 2021b; Flet-Berliac et al., 2021).

In all cases, we seek to train a student agent via Proximal Policy Optimization (PPO, Schulman et al., 2016-2018), with a ResNet policy (He et al., 2016) as originally proposed in Espeholt et al. (2018). To evaluate the quality of the curricula, we show all performance with respect to the number of student gradient updates as opposed to total environment interactions, which is often comparable for PLR and ACCEL. For a full list of hyperparameters for each experiment please see Table 6 in Section B.3. As baselines we consider the following:

- **Domain Randomization (DR)**: Randomly sampling from a parameterized distribution.
- **Prioritized Level Replay (PLR)**: We use Robust PLR from Jiang et al. (2021a).
- **PAIRED**: Placing a fixed number of blocks, using the algorithm described in Dennis et al. (2020).
- **Minimax Adversarial**: Placing blocks using an adversarial objective seeking to minimize return.

As in Dennis et al. (2020), we use a single minimax represent the POET objective when it is not combined with hand-coded constraints on the levels generated. We leave the comparison to population-based methods to future work due to the computational expense.

### 4.1 LEARNING WITH LAVA

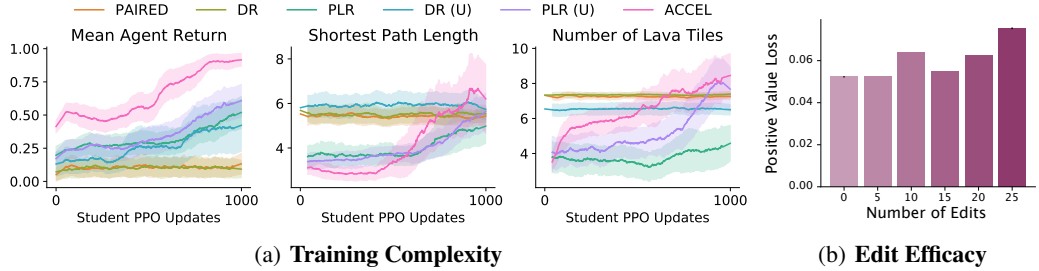

(a) **Training Complexity**  (b) **Edit Efficacy**

Figure 4: Lava grid training data. a) Left to right: mean agent return on training levels, the shortest path length and the number of lava tiles, plots show mean and sem. b) Positive Value Loss by number of edits for ACCEL.

We begin with a simple grid environment, whereby the agent must navigate to a goal in the presence of lava blocks. The grid is small, only 7x7, however, it is challenging for RL agents since exploring with random actions often leads to instant death, which makes it unlikely to receive a signal from the sparse reward. Indeed, this makes the choice of DR parameterization crucial, since sampling too many blocks early on will be prohibitive for learning. We compare two different parameterizations: "Binomial" where the agent samples from {lava, none} for 20 steps, and "Uniform" where the agent samples the number of tiles to place from the range [0,20]. For ACCEL, we use a generator that

produces empty rooms and then proceed to edit the levels to add (or remove) lava blocks. The environment is built with MiniHack, thus the agent has a global observation (details in the Appendix, Section: B.1). We ran each method for five seeds, showing the results in Fig 4.

As we see, ACCEL quickly produces levels with more lava than the other methods, while also getting near-perfect return on its training distribution. Interestingly, with the Uniform parameterization, we see that PLR is able to produce a similar training profile to ACCEL, but achieves a lower value in every individual metric. The methods with learned generators (PAIRED and minimax) fail to learn anything, thus are unable to form a curriculum (Figure 4). Finally on the right we took a snapshot of the level replay buffer for ACCEL, where we clearly see that the levels which have been edited more have higher approximate regret. After one thousand PPO updates (around 20M timesteps) we tested each agent on a series of test tasks, which we show in the Appendix (see Section A.3). We thus have answered our first question: in a design space with a high proportion of challenging levels, ACCEL is able to build an effective curriculum which quickly facilitates learning on the full distribution.

## 4.2 PARTIALLY OBSERVABLE NAVIGATION

To answer the second question, we now scale to the MiniGrid (Chevalier-Boisvert et al., 2018) setting originally introduced in Dennis et al. (2020). Despite being a conceptually simple environment, this is a large experiment: our agents train for 20k updates (around 350M steps, see Table 5), learning an LSTM-based policy with a 147 dimension partially observable observation. We use the Uniform parameterization for DR which first samples the number of blocks to place, ranging from zero (an empty room) to sixty, since previous works showed that DR is sensitive to the number of blocks placed (Jiang et al., 2021a). For ACCEL we begin with empty rooms and randomly edit the block locations (adding or removing them) as well of the goal location. After replay, we edit the "easy" levels, essentially moving levels back to the frontier once their learning potential has been reduced.

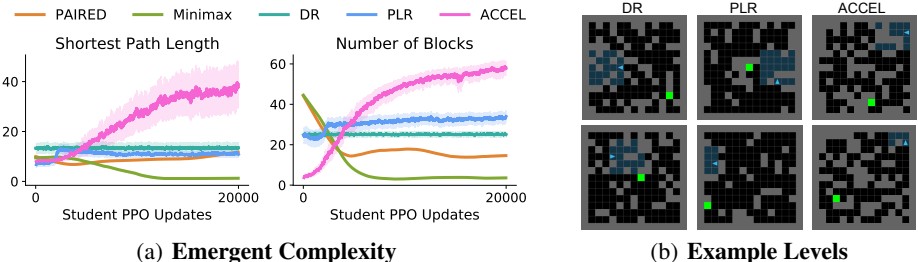

| (a) **Emergent Complexity** | (b) **Example Levels** |

Figure 5: MiniGrid training data. a): the performance of all agents on the the shortest path length and number of blocks on training levels, where ACCEL quickly develops highly challenging levels. Plots show the mean and standard error across five runs. b) Example levels generated by DR, PLR and ACCEL.

In Figure 5.a) we show the training performance, where ACCEL does exactly as intended—it compounds initial complexity to ultimately train on levels with high block count and long paths to the goal. This can be seen in 5.b), where ACCEL produces more structured mazes than the baselines. We evaluate all five methods zero-shot on a series of held-out environments as used in prior works (see Figure 16), with the mean and sem per environment shown in Figure 6. For DR, PLR and ACCEL the evaluation is after 20k student updates, thus it solely compares the quality of the curriculum, while we use the Minimax and PAIRED results from Jiang et al. (2021a) at 250M training steps (>30k updates). As we see, ACCEL performs as least as well as the next best method in almost all settings, with particular strength in the more complex Labyrinth and Maze environments.

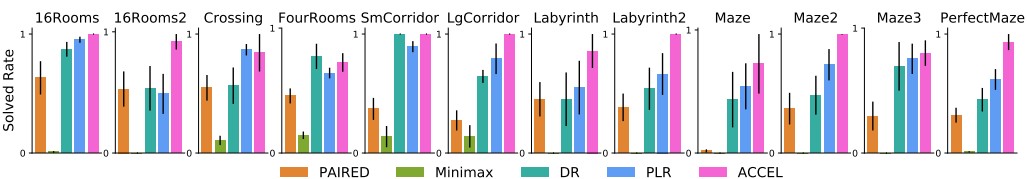

Figure 6: Zero-shot transfer results. Agents are evaluated for 100 episodes on a series of human designed mazes, plots show mean and standard error for each environment, across five runs.

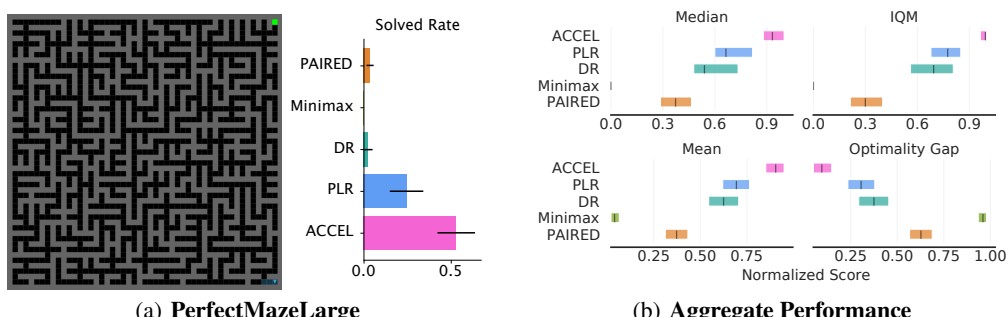

(a) **PerfectMazeLarge**

(b) **Aggregate Performance**

Figure 7: a) Zero-shot performance on a large procedurally-generated maze environment. Agents are evaluated for 100 episodes, bars show mean and standard error. ACCEL achieves over double the success rate of the next best method, despite beginning with empty rooms. b) Example levels produced by each UED algorithm.

To evaluate the performance at the aggregate level, we make use of the recently introduced `rliable` library (Agarwal et al., 2021b) in Figure 7.b). At the aggregate level the strength of ACCEL is clear, with an IQM[2] near 100% solved rate compared to below 80% for PLR, with a probability of improvement of 80.2%. It is clear that ACCEL is significantly stronger in these test environments.

Next we consider an even more challenging setting—we use a larger version of the "PerfectMaze", a procedurally-generated maze environment, shown in Figure 7.a). The maze has 51x51 tiles, an order of magnitude larger than the training environment, and has a maximum episode length of over 5k steps. This is a daunting navigation challenge, requiring extensive use of memory so as not to get lost in a loop of repeatedly exploring the same paths. We evaluate the agents at the checkpoint from Figure 6, testing each seed for 100 episodes, showing the mean and standard error in Figure 7.a). Both versions of ACCEL significantly outperform all baselines, achieving success rates of 53% and 52% compared to the next best 25% for PLR, while all other methods fail. Notably, ACCEL appears to approximately follow the "left-hand" rule for solving mazes.[3]

**What if we edit DR levels?** We also consider an ablation of ACCEL where instead of beginning with empty rooms, we instead begin with levels sampled from the DR distribution. In Figure 8 we show the performance of ACCEL, DR and PLR on the same generator distribution during training, where we see that ACCEL outperforms both. We also tested this version of ACCEL on the same held out tasks, where it achieved a mean performance comparable to the version presented in Figure 6, while it also achieved 52% success on the large perfect maze. We conducted other ablations such as using a learned editor, or editing the full batch, with only small changes in performance (see Section A.4). We believe this shows the strength of ACCEL, that it is robust to a multitude of factors, even the generator distribution.

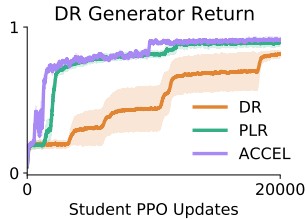

Figure 8: Performance of ACCEL when using the Uniform DR generator, compared to DR and PLR. We test all agents during training on the same DR distribution. Curves show mean with sem shaded.

### 4.3 DISCUSSION AND LIMITATIONS

In our experiments we have demonstrated that ACCEL is capable of forming highly effective curricula in two challenging navigation environments. In the first, we showed ACCEL can facilitate learning in a design space with a high proportion of hard levels, which could have an impact in improving exploration in safety-critical settings. In the second, we showed it is possible to produce complex mazes which facilitate zero-shot transfer to human-designed ones, scaling to environments an order of magnitude larger than the training environment. This is made possible since ACCEL produces a high frequency of solvable mazes with a high block count, regardless of the DR generator used. Note that PLR is capable of sampling 60 block levels, but it will infrequently sample those that also contain a useful (solvable) path to the goal. We thus believe we have shown evidence that ACCEL would be an effective method for training agents in more open-ended UED design spaces.

---

[2]Interquartile Mean (IQM) is the recommended robust statistic in Agarwal et al. (2021b).

[3]For more details see `https://sites.google.com/view/compoundingcomplexity`

However, as with any method, ACCEL comes with limitations. Our approach includes an inductive bias with the ability to begin with a simple base case (an empty room). This may not always be possible in practice, while in some settings the simplest example (in terms of entities placed in the environment) may actually be a more difficult environment to solve (for example in a Hide and Seek game). In addition, our experiments thus far only consider navigation tasks, and while the MiniGrid experiments are one of the largest settings used in UED, for the field more broadly to become useful for real-world problems it will be necessary to test new environments. On the algorithmic side, a potential limitation of ACCEL is there may be a lack of diversity in the level replay buffer. In this work we do not explicitly optimize for diversity, but instead, seek to reduce the impact through level replay hyperparameters (see the Appendix, Figure 21). However, scaling ACCEL further may require a mechanism to directly encourage diversity.

## 5 RELATED WORK

Our work straddles a variety of interrelated fields, which we discuss in this section. For a summary of the most closely related methods see Table 1. Our paper focuses on testing agents on distributions of environments, which has long been known to be crucial to evaluate the generalization capability of RL agents (Whiteson et al., 2009). The failure of agents in this setting has recently drawn considerable attention (Packer et al., 2019; Igl et al., 2019; Cobbe et al., 2020; Agarwal et al., 2021a; Zhang et al., 2018b; Ghosh et al., 2021), with policies often failing to adapt to changes in the observation (Song et al., 2020), dynamics (Ball et al., 2021) or reward (Zhang et al., 2018a). In this work, we seek to provide agents with a set of training levels to produce a policy that is robust to these variations.

In particular, we focus on the *Unsupervised Environment Design* (UED, Dennis et al., 2020) paradigm, which shifts from designing agents that can generalize from a fixed distribution of environments towards designing the environments themselves. The most popular method for UED is Domain randomization (DR, Jakobi, 1997; Sadeghi & Levine, 2017) which has been particularly successful in areas such as robotics (Tobin et al., 2017; James et al., 2017; Andrychowicz et al., 2020; OpenAI et al., 2019), with extensions proposing to actively update the DR distribution Mehta et al. (2020); Raparthy et al. (2020). This paper directly extends *Prioritized Level Replay* (PLR, Jiang et al., 2021b;a), a method for curating DR levels such that those with high learning potential can be replayed a student agent. PLR is related to TSCL (Matiisen et al., 2020), self-paced learning (Klink et al., 2019; Eimer et al., 2021) and ALP-GMM Portelas et al. (2019), which seek to maintain and update distributions over environment parameterizations. Very recently it was shown that with a smooth task space, a method similar to PLR is capable of producing generally capable agents in a simulated game world (Team et al., 2021), using large scale compute and Population Based Training (Jaderberg et al., 2017). However, this work relied on a highly optimized task design space, which is rarely present in practice.

Dennis et al. (2020) introduced the PAIRED algorithm, an elegant approach for UED, whereby an environment adversary optimizes for *minimax regret* (Savage, 1951), defined as the difference in performance between an antagonist agent (colluding with the adversary) and the protagonist. This guarantees the adversary produces solvable mazes, which allows the protagonist to transfer to unseen environments and even learn to navigate the web (Gur et al., 2021). Adversarial objectives have also been considered in robotics (Pinto et al., 2017). POET (Wang et al., 2019; 2020) considers evolving a *population* of environments, each paired with an agent, using an objective similar to minimax adversarial, which needs to be combined with domain-specific rules to prevent unsolvable environments from being proposed. We take inspiration from the evolutionary nature of POET but train a *single agent*, which is beneficial as it takes significantly fewer resources, while also removing the agent selection problem. UED is inherently related to the field of *Automatic Curriculum Learning* (ACL, Portelas et al., 2020; Florensa et al., 2017; Baranes & Oudeyer, 2009), which seeks to provide a curriculum of increasingly challenging tasks or goals given a (typically) fixed environment. A canonical approach is Hindsight Experience Replay (Andrychowicz et al., 2017) which was shown to be effective for sparse reward tasks. Asymmetric Self Play (Sukhbaatar et al., 2018) takes the form of one agent proposing goals for another, which was shown to be effective for challenging robotic manipulation tasks (OpenAI et al., 2021). AMIGo (Campero et al., 2021) and APT-Gen (Fang et al., 2021) provide solutions to problems where the target task is known, providing a curriculum of increasing difficulty. Indeed, many ACL methods emphasize learning to reach goals or states with high uncertainty (Racaniere et al., 2020; Pong et al., 2020; Zhang et al., 2020), either using generative (Florensa et al., 2018) or world models (Mendonca et al., 2021). Unlike these

Table 1: The components of related approaches. Like POET, we evolve levels, but use a single agent rather than a population, while also using a minimax regret objective, which ensures the environments generated are solvable. PAIRED uses minimax regret for the generator, which is often challenging to optimize, while it does not replay levels so may suffer from cycling. Finally, PLR curates levels using minimax regret, but relies solely on domain randomization for generation.

| Algorithm | Generation Strategy | Generator Obj | Curation Obj | Setting |
|---|---|---|---|---|
| POET (Wang et al., 2019) | Evolution | Minimax | MCC | Population-Based |
| PAIRED (Dennis et al., 2020) | Reinforcement Learning | Minimax Regret | None | Single Agent |
| PLR (Jiang et al., 2021b;a) | Random | None | Minimax Regret | Single Agent |
| ACCEL | Random + Evolution | Minimax Regret | Minimax Regret | Single Agent |

methods, UED approaches seek to fully specify environments, rather than just goals within a fixed environment.

In the symbolic AI community, *environment design* has been considered as a means to alter an environment to influence an agent's decisions (Zhang & Parkes, 2008; Zhang et al., 2009). This was extended with automated design (Keren et al., 2017; 2019), which seeks to redesign environments given the limitations of agents, to improve their performance. Unlike these works, ACCEL seeks to automatically design environments in order to produce a curriculum for a learned agent.

Our work also closely relates to the field of *Procedural Content Generation* (PCG, Risi & Togelius, 2020; Justesen et al., 2018), where levels are sampled from a distribution. Popular PCG settings include the Procgen Benchmark (Cobbe et al., 2020), MiniGrid (Chevalier-Boisvert et al., 2018), Obstacle Tower (Juliani et al., 2019), GVGAI (Perez-Liebana et al., 2019) and the NetHack Learning Environment (Küttler et al., 2020). This work uses the recently proposed MiniHack environment (Samvelyan et al., 2021), which provides a flexible framework for creating diverse environments. Within the PCG community, automatically generating game levels has been of interest for more than a decade (Togelius & Schmidhuber, 2008). More recently, machine learning has proven to be effective (Summerville et al., 2018; Bhaumik et al., 2020; Liu et al., 2021). Related to our work, PCGRL (Khalifa et al., 2020; Earle et al., 2021a) framed level design as an RL problem, designing environments by making incremental changes. However, it makes use of hand-designed dense rewards, and focuses on the design of levels for *human* players. By contrast, ACCEL seeks to train a student agent and does not require domain-specific feedback.

## 6 Conclusion and Future Work

In this paper we proposed a new method for Unsupervised Environment Design (UED), ACCEL, which evolves a curriculum by *editing* previously curated levels. This makes it possible to constantly generate a wide variety of environments at the frontier of the agent's capabilities, producing curricula that start simple and quickly compound. We believe ACCEL offers the best of both worlds: a principled regret-based curriculum that does not require domain-specific heuristics, alongside an evolutionary process that produces a broad spectrum of complexity catering to the agent's current capabilities. In our experiments we showed that ACCEL is capable of efficiently training agents that can transfer to a series of human-designed environments, outperforming competitive baselines.

For future work, it may be possible to use ideas from Neural Cellular Automata to enhance the editing process (Earle et al., 2021b), possibly making use of controllable editors that can boost specific properties of levels (Earle et al., 2021a). We could also consider generative modelling approaches to level editing, training on PLR levels to produce new frontier levels. We did not explore methods to encourage levels to be diverse, but this would likely be important for larger-scale experiments. Another possibility is to actively seek levels which have high "Evolvability" (Gajewski et al., 2019). This could be increased by introducing so-called "extinction events" (Raup, 1986), which have been shown to increase Evolvability (Lehman & Miikkulainen, 2015) and are believed to play a crucial role in natural evolution. Equipped with some of these ideas, we could go significantly further towards open-endedness (Earle et al., 2021c). We believe it may be possible to increase the search space such that MiniHack can make broader use of the richness of the NetHack world, possibly aiding us in making progress in the full game of Nethack—a grand challenge in reinforcement learning.

STATEMENT

We do not perceive any potential ethical issues to arise from this work, beyond those typically associated with reinforcement learning. However, a key issue in RL is reproducibility, and as such we will be open sourcing our code alongside the camera ready version of our paper. Both environments we used are open sourced (MiniGrid and MiniHack) and we based our code on an open source repo.

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

# A    ADDITIONAL EXPERIMENTAL RESULTS

## A.1    LEVEL EVOLUTION

In Fig 9 and 10 and we show additional levels produced by ACCEL for the MiniHack lava environment and MiniGrid mazes respectively. Note that in all cases, each incremental step along the evolutionary process produces a level that has high learning potential *at that point in time*.

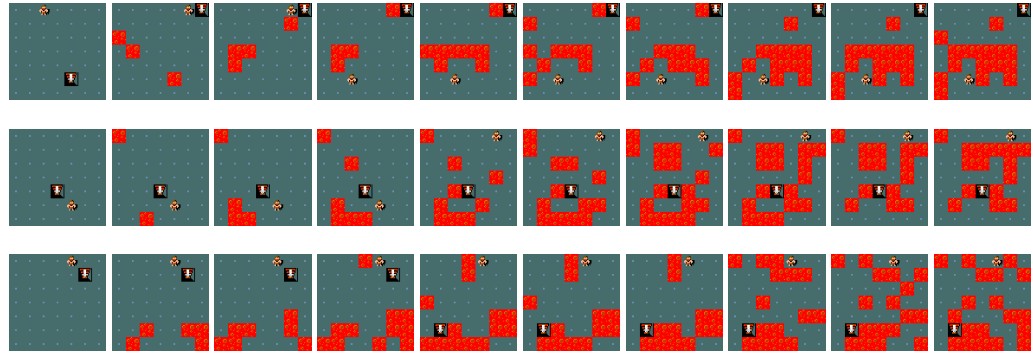

Figure 9: Levels generated by ACCEL. Note that in all cases, each individual level along the evolutionary path is at the frontier for the student agent at that stage of training. As we can see, the edits compound to produce a series of challenges: in the first level the lava gradually surrounds the agent, such that they can initially explore in multiple directions but at the end the task can only be solved by going down and to the right. In the middle row we see a level where the agent always has a direct run at the goal, but a corridor is evolved over time to become increasingly narrow, before being filled in so the agent has to go around. Finally in the bottom row the level begins with simple augmentations before moving the agent behind a barrier, which results in a challenging task where the agent has to move in a diagonal direction to escape the lava.

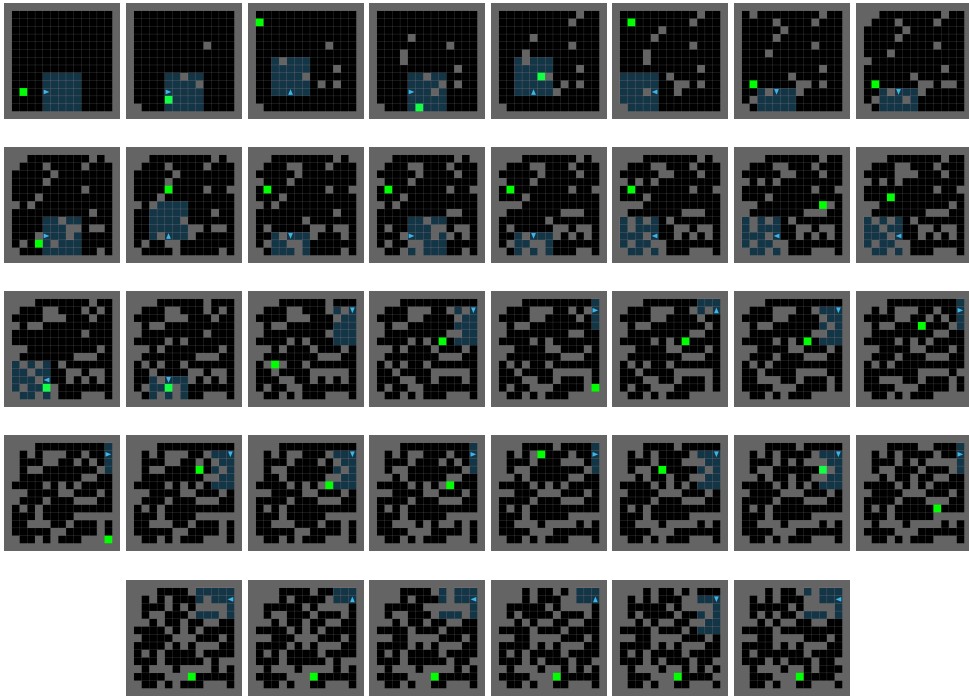

Figure 10: A single level evolved in the MiniGrid environment, starting from top left, ending bottom right. Throughout the process the agent experiences a diverse set of challenges.

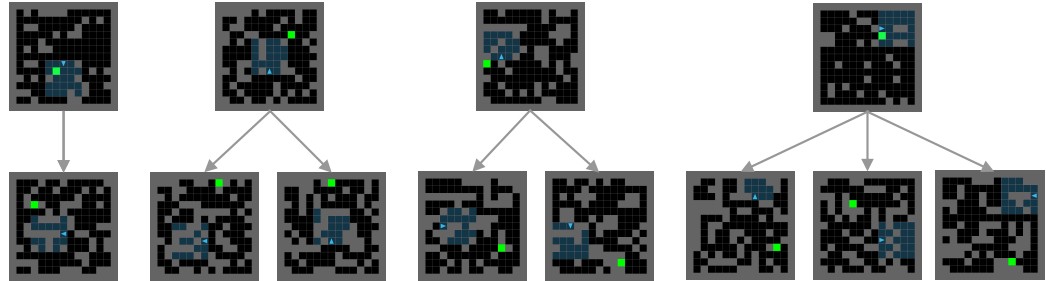

Figure 11: Maze evolution with the DR generator. Top row shows starting DR levels, originally included in the replay buffer due to having high positive value loss. After many edits (up to 40), they produce the bottom row, which were all chosen to be in the highest 50 replay scores after 10k gradient steps. As we see, the same DR level can produce distinct future levels, in some cases multiple high regret levels.

## A.2 THE EXPANDING FRONTIER

Here we analyze the performance of agents on levels produced by ACCEL. We have four agent checkpoints, from 5k, 10k, 15k and 20k student gradient updates. In Figure 12 we show four generations of a level, where we see that the later generations become harder for the 5k checkpoint, while the 20k checkpoint gets the highest learning signal (Positive Value Loss) from Generation 63.

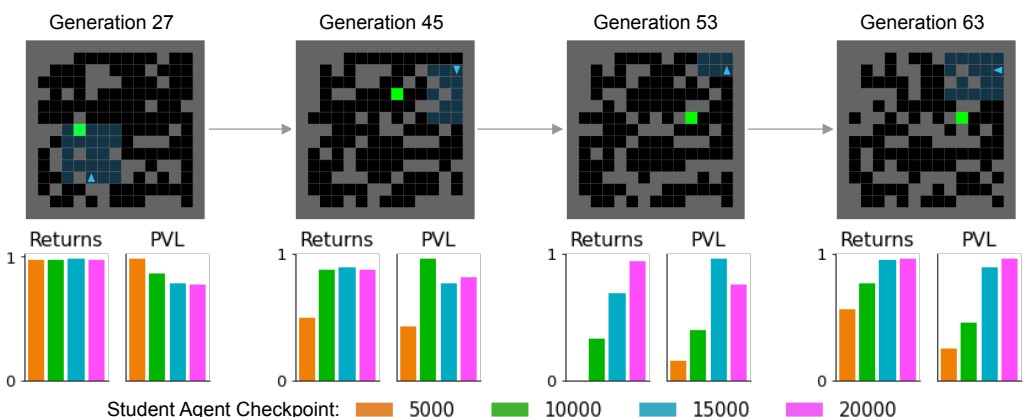

Figure 12: The Evolving Frontier. The top row shows four levels from the same lineage, at generations 27, 45, 53 and 63. Underneath each is a bar plot showing the Return and Positive Value Loss (PVL) for four different ACCEL policies, saved after 5k, 10k, 15k and 20k updates. At generation 27, all four checkpoints can solve the level, but the 5k checkpoint has the highest learning potential (PVL). On the right we see that by generation 63, only the 15k and 20k checkpoints are able to achieve a high return on the level.

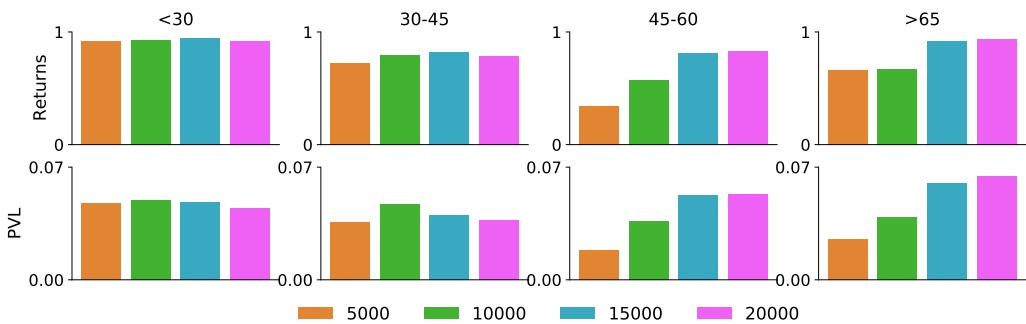

Figure 13: Aggregate metrics for each band of generations. For example, "30-45" refers to all the levels between generation 30-45. The later generation levels are harder for the early agents to solve, while the early agents have higher return and PVL for the earlier levels.

In Figure 13 we show all generations for the level included in Figure 12, grouped by generation. We then show the mean Return and Positive Value loss (PVL) for all four agent checkpoints. It is clear to see that the later generations have the highest learning potential for the 20k checkpoint, with the lowest return for the 5k checkpoint.

Next in Figure 14 we show data for all generations of 20 levels, chosen as those which were present in the 20k checkpoint replay buffer but also had ancestors in the 5k checkpoint buffer. For each checkpoint we plot all levels along the dimensions of Number of Blocks and Shortest Path, with the color corresponding to the solved rate. On the left, the 5k checkpoint agent can only repeatedly solve the shorter path levels with low block count. Later on, the 20k checkpoint agent performs well across the entire space.

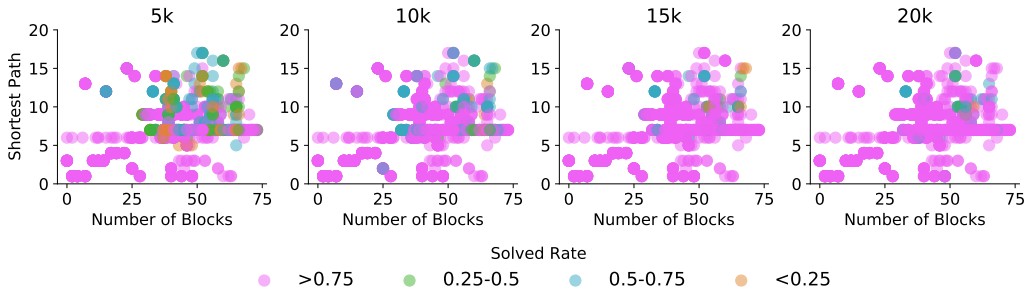

Figure 14: How do complexity metrics relate to difficulty? The plot shows the block count and shortest path length. From left to right we evaluate the agents at four checkpoints: 5k, 10k, 15k, 20k PPO updates. The color represents the solved rate. As we see, the 5k agent is unable to solve the levels with higher block count and longer paths to the goal, while the 20k agent is able to solve almost all levels.

## A.3 FULL EXPERIMENTAL RESULTS

**Lava Grid Extended Results** In Table 2 and Figure 15 we include the full results for the MiniHack lava experiments. The first three (Empty, 10 Tiles and 20 Tiles) evaluate the performance of the agent within its training distribution, while we also include a held-out human designed environment, LavaCrossing S9N1, ported from MiniGrid Chevalier-Boisvert et al. (2018). As we see, ACCEL performs best on all of the in sample environments, while also being only one of two approaches to get meaningfully above zero in the human designed task.

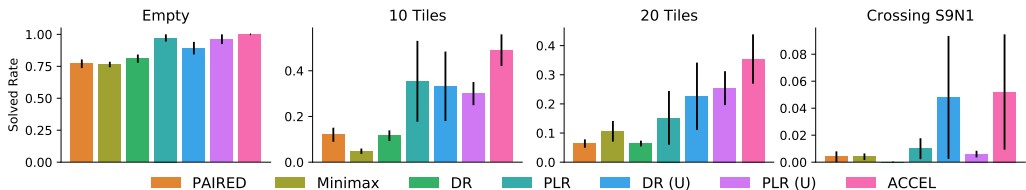

Figure 15: Test performance, both in distribution (Empty, 10 and 20 Tiles) and out of distribution (Crossing S9N1). Each evaluation is conducted for 100 trials. Plots show the mean and standard error across five runs.

Table 2: Test performance in four environments. Each data point corresponds to the mean (and standard error) of five independent runs, where each run is evaluated for 100 trials on each environment. † indicates the generator distribution is a Binomial, whereby the generator can place 20 blocks, each is either a lava tile or empty. ‡ indicates the generator first samples the number of lava tiles to place, between zero and 20, then places that many. Bold indicates being within one standard error of the best mean.

| Test Environment | PAIRED | Minimax | DR† | PLR† | DR‡ | PLR‡ | ACCEL |
|---|---|---|---|---|---|---|---|
| Empty | $0.77 \pm 0.03$ | $0.76 \pm 0.02$ | $0.81 \pm 0.03$ | $0.97 \pm 0.03$ | $0.89 \pm 0.05$ | $0.96 \pm 0.04$ | $\mathbf{1.0 \pm 0.0}$ |
| 10 Tiles | $0.12 \pm 0.03$ | $0.05 \pm 0.01$ | $0.12 \pm 0.02$ | $0.35 \pm 0.18$ | $0.33 \pm 0.15$ | $0.3 \pm 0.05$ | $\mathbf{0.49 \pm 0.07}$ |
| 20 Tiles | $0.06 \pm 0.01$ | $0.11 \pm 0.04$ | $0.06 \pm 0.01$ | $0.15 \pm 0.09$ | $0.23 \pm 0.12$ | $0.25 \pm 0.06$ | $\mathbf{0.35 \pm 0.08}$ |
| LavaCrossing S9N1 | $0.0 \pm 0.0$ | $0.0 \pm 0.0$ | $0.0 \pm 0.0$ | $0.01 \pm 0.01$ | $\mathbf{0.05 \pm 0.05}$ | $0.01 \pm 0.0$ | $\mathbf{0.05 \pm 0.04}$ |

**Partially-Observable Navigation** Next we show the extended results for the MiniGrid experiments. We use a series of challenging zero-shot environments (see Figure 16), introduced in the UED literature (Dennis et al., 2020; Jiang et al., 2021a). We include the full results in Table 3.

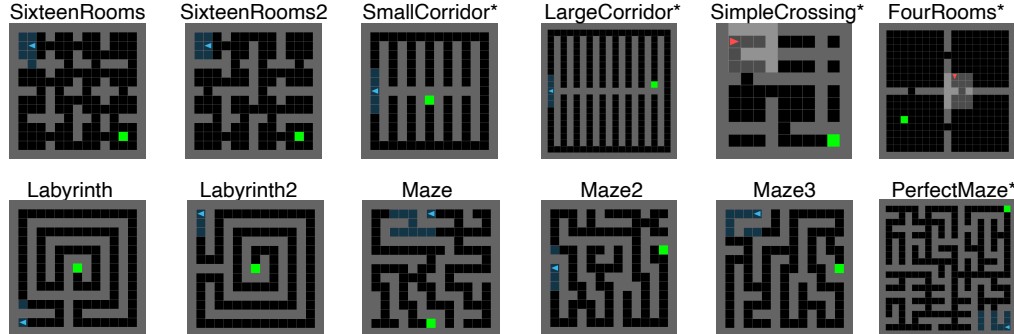

Figure 16: MiniGrid Zero-Shot Environments. Those with an asterisk are procedurally generated: For Small and Large Corridor, the position of the goal can be in any of the corridors, for SimpleCrossing and Four Rooms see Chevalier-Boisvert et al. (2018) and for PerfectMaze see Jiang et al. (2021a).

Table 3: Zero-Shot transfer to human-designed environments. Each data point corresponds to the mean (and standard error) of five independent runs, where each run is evaluated for 100 trials on each environment. † indicates the generator first samples the number of blocks to place, between zero and sixty, then places that many. ‡ indicates the generator produces empty rooms. Bold indicates being within one standard error of the best mean. ⋆ indicates $p < 0.05$ in Welch's t-test against PLR. Note that all methods are evaluated after 20k student updates, aside from PAIRED and Minimax which have 30k updates.

| Environment | PAIRED | Minimax | DR† | PLR† | ACCEL† | ACCEL‡ |
|---|---|---|---|---|---|---|
| 16Rooms | $0.63 \pm 0.14$ | $0.01 \pm 0.01$ | $0.87 \pm 0.06$ | $0.95 \pm 0.03$ | $\mathbf{1.0 \pm 0.0}$ | $\mathbf{1.0 \pm 0.0}$ |
| 16Rooms2 | $0.53 \pm 0.15$ | $0.0 \pm 0.0$ | $0.53 \pm 0.18$ | $0.49 \pm 0.17$ | $0.62 \pm 0.22$ | $\mathbf{0.92 \pm 0.06}$ |
| SimpleCrossing | $0.55 \pm 0.11$ | $0.11 \pm 0.04$ | $0.57 \pm 0.15$ | $\mathbf{0.87 \pm 0.05}$ | $\mathbf{0.92 \pm 0.08}$ | $\mathbf{0.84 \pm 0.16}$ |
| FourRooms | $0.46 \pm 0.06$ | $0.14 \pm 0.03$ | $0.77 \pm 0.1$ | $0.64 \pm 0.04$ | $\mathbf{0.9 \pm 0.08}$ | $0.72 \pm 0.07$ |
| SmallCorridor | $0.37 \pm 0.09$ | $0.14 \pm 0.09$ | $\mathbf{1.0 \pm 0.0}$ | $0.89 \pm 0.05$ | $0.88 \pm 0.11$ | $\mathbf{1.0 \pm 0.0}$ |
| LargeCorridor | $0.27 \pm 0.08$ | $0.14 \pm 0.09$ | $0.64 \pm 0.05$ | $0.79 \pm 0.13$ | $0.94 \pm 0.05$ | $\mathbf{1.0 \pm 0.0}$ |
| Labyrinth | $0.45 \pm 0.14$ | $0.0 \pm 0.0$ | $0.45 \pm 0.23$ | $0.55 \pm 0.23$ | $\mathbf{0.97 \pm 0.03}$ | $0.86 \pm 0.14$ |
| Labyrinth2 | $0.38 \pm 0.12$ | $0.0 \pm 0.0$ | $0.54 \pm 0.18$ | $0.66 \pm 0.18$ | $\mathbf{1.0 \pm 0.01}$ | $\mathbf{1.0 \pm 0.0}$ |
| Maze | $0.02 \pm 0.01$ | $0.0 \pm 0.0$ | $0.43 \pm 0.23$ | $\mathbf{0.54 \pm 0.19}$ | $0.52 \pm 0.26$ | $\mathbf{0.72 \pm 0.24}$ |
| Maze2 | $0.37 \pm 0.13$ | $0.0 \pm 0.0$ | $0.49 \pm 0.16$ | $0.74 \pm 0.13$ | $0.93 \pm 0.04$ | $\mathbf{1.0 \pm 0.0}$ |
| Maze3 | $0.3 \pm 0.12$ | $0.0 \pm 0.0$ | $0.69 \pm 0.19$ | $0.75 \pm 0.12$ | $\mathbf{0.94 \pm 0.06}$ | $0.8 \pm 0.1$ |
| PerfectMaze (M) | $0.32 \pm 0.06$ | $0.01 \pm 0.0$ | $0.45 \pm 0.1$ | $0.62 \pm 0.09$ | $\mathbf{0.88 \pm 0.12}$ | $\mathbf{0.93 \pm 0.07}$ |
| **Mean** | $0.39 \pm 0.03$ | $0.05 \pm 0.01$ | $0.62 \pm 0.05$ | $0.71 \pm 0.04$ | $\mathbf{0.88 \pm 0.04^{\star}}$ | $\mathbf{0.9 \pm 0.03^{\star}}$ |

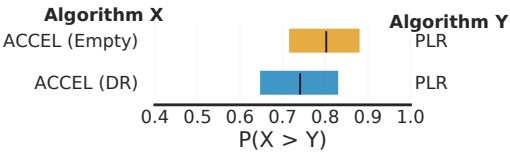

Figure 17: Probability of improvement of ACCEL vs. PLR across all the benchmark environments in Figure 16, using the open source notebook fromm Agarwal et al. (2021b). The probability of improvement represents the probability that Algorithm X outperforms Algorithm Y on a new task from the same distribution.

As we see, both versions of ACCEL significantly outperform the baselines. Particularly in the more complex environments like Labyrinth we see large gains vs. the baselines. Also note that PLR outperforms all other baselines, and ACCEL outperforms PLR. We show this in the Table with a statistically significant point estimate, but also using the more robust metrics in `rliable` Agarwal et al. (2021b), for example the "probability of improvement" shown in Figure 17.

**Testing the Limits of Current Approaches** In the experimental section we showed the results for a large procedurally generated maze, of size 51x51. ACCEL averaged a 53% and 52% success rate when using the empty or DR generator respectively, vs. a next best PLR with 25%. Now we consider going even further. We tested both versions of ACCEL, as well as DR and PLR, on an even larger maze, this time 101x101, shown in Figure 18. Note that this would be challenging even for human players, since the agent has a partially observable view. The performance of all methods is significantly weaker, with DR and PLR achieving a mean of 4% success rate. However, ACCEL still outperforms, achieving 7% and 8% success rate with the DR and empty generators respectively. We believe at this point the bottleneck for further improvement may not be the curriculum, but instead the LSTM-based policy which has to remember all previous paths, to solve the maze in the allotted 20,402 steps.

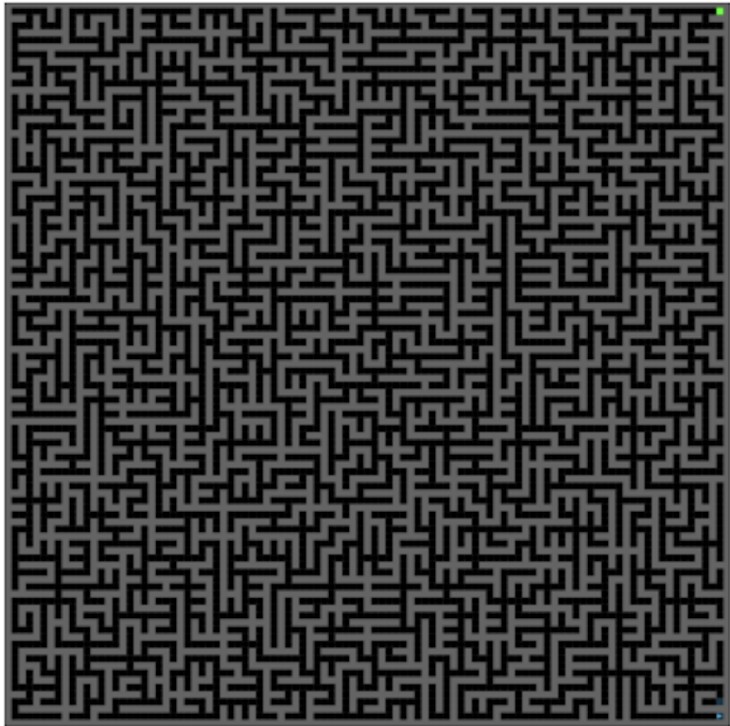

Figure 18: PerfectMazesXL. A 101x101 procedurally-generated MiniGrid environment. The agents have to transfer zero-shot from training in a 15x15 grid. This environment is challenging even for humans, since the agent only has a partially observable view, it requires memorizing the current location at all times to ensure exploring all corners of the grid.

## A.4 ADDITIONAL EXPERIMENTS

In this section we show a series of additional experiments to understand the performance of ACCEL.

**The Sensitivity of Domain Randomization** Consider a simple parameterized grid world environment where an agent has to navigate towards a goal. Each level varies in terms of the location of the agent, goal and placement of obstacles. When these obstacles are *walls* the agent is able to learn quickly, achieving high performance on the training distribution. However, when these obstacles are *lava*, which causes instant death on collision, the agent is unable to learn (Figure 19.a), left). If we change the DR parameterization to not only sample the locations of the lava tiles, but the *number of tiles to place*, then the agent does indeed learn (Figure 19.a), right). This subtle change illustrates the sensitivity of DR to the environment parameterization. We refer to this problem as "curse of dimensionality for UED", whereby increasing the design space leads to a crowding out of levels that are effective for learning.

In Figure 19.b) we show the test performance for the three DR agents at the same checkpoint (1000 updates), as well as a PLR agent using the Binomial DR levels. Despite the poor performance for the

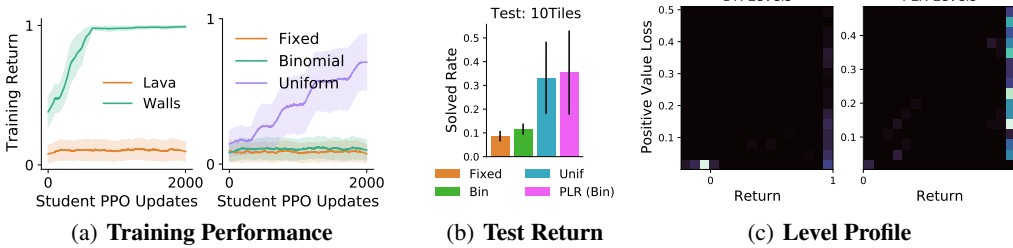

(a) **Training Performance**  (b) **Test Return**  (c) **Level Profile**

Figure 19: Small grid results: a) On the left, we compare training with DR, randomizing obstacles that can be walls or lava, on the right we see that when the obstacles are lava the choice of DR parameterization has a significant impact on performance. In b) we see the test performance of these agents after 1k updates, PLR applied to the Binomial (Bin) levels is able to produce a strong agent, why? In c) we see a density plot of DR and PLR levels, showing frequency by Return and Positive Value Loss, which shows PLR curates solvable levels where the agent receives a learning signal.

DR agent, PLR is able to achieve strong performance, outperforming even the Uniform DR agent. To see how this is possible, in Figure 19.c) we compare the density of levels sampled by DR vs. those in the PLR buffer, sorted by Return and Positive Value Loss. Notably, the levels chosen by PLR are inherently higher return levels, thus, we see that PLR is able to find the frontier without handcrafted heuristics. This example illustrates the importance of having a sufficient quantity of frontier levels. Indeed, the success of PLR hinges on the ability to reliably sample new levels at the ever-changing frontier of the agent's capabilities. This is the key motivation for ACCEL.

Next, in Figure 20 we show the performance of the three DR agents when tested on each other's training distributions after one thousand training steps. All agents perform worst on the Fixed levels, slightly better on the Binomial ones, with the best performance on Uniform. Notably, even the Fixed agent, the weakest of the three, is able to see almost double the performance on the Uniform levels. This indicates that a key ingredient to learning potential is having a sufficient number of easier levels to generate learning signal. Indeed, this was noted in Jiang et al. (2021b) in their MiniGrid experiments.

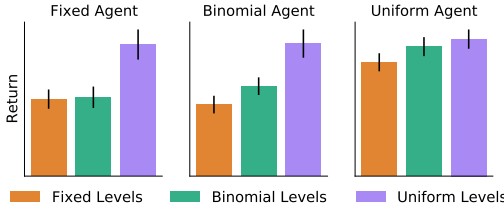

Figure 20: Performance of agents trained on different level distributions when tested on each of the distributions. It is clear to see the Uniform distribution contains more solvable levels—all three agents perform better on it.

**Ablation Studies** To investigate the design choices that led to the success of ACCEL, we consider a variety of ablations:

- **ACCEL** Using the DR generator.
- **Edit Batch** The same ACCEL algorithm but editing the entire replay batch rather than the "easy" levels from the batch.
- **Learned Editor** The same algorithm changing only the editor, which is optimized for positive value loss.
- **No Editor** This is an ablation on the algorithm structure, we use Algorithm 1 but sample more DR levels instead of editing.

Each of these uses the same exact structure, sampling levels from the DR distribution 10% of the time, and editing levels after replaying from the curator. For the first, we edit the entire replay batch, rather than just the top four easiest (high return minus positive value loss) that we use in the main experiments. The results after 10k updates are shown in Table4. As we see, Editing the batch levels performs slightly worse than easy, while there is also a decrease in performance for the learned editor.

Table 4: Zero-Shot transfer to human-designed environments. Each data point corresponds to the mean (and standard error) of five independent runs, where each run is evaluated for 100 trials on each environment. All methods use a DR generator which places between zero and sixty blocks.

| Test Environment | ACCEL | Edit Batch | Learned Editor | No Editor |
|---|---|---|---|---|
| 16Rooms | $1.0 \pm 0.0$ | $0.76 \pm 0.19$ | $0.9 \pm 0.07$ | $0.84 \pm 0.06$ |
| 16Rooms2 | $0.51 \pm 0.28$ | $0.23 \pm 0.16$ | $0.41 \pm 0.19$ | $0.68 \pm 0.18$ |
| SimpleCrossing | $0.8 \pm 0.05$ | $1.0 \pm 0.0$ | $0.9 \pm 0.1$ | $0.75 \pm 0.05$ |
| FourRooms | $0.85 \pm 0.05$ | $0.85 \pm 0.06$ | $0.88 \pm 0.04$ | $0.88 \pm 0.05$ |
| SmallCorridor | $0.72 \pm 0.1$ | $0.74 \pm 0.1$ | $0.6 \pm 0.17$ | $0.7 \pm 0.18$ |
| LargeCorridor | $0.91 \pm 0.05$ | $0.75 \pm 0.08$ | $0.56 \pm 0.18$ | $0.63 \pm 0.18$ |
| Labyrinth | $0.98 \pm 0.02$ | $0.85 \pm 0.11$ | $0.99 \pm 0.01$ | $0.67 \pm 0.19$ |
| Labyrinth2 | $0.97 \pm 0.03$ | $0.83 \pm 0.11$ | $0.7 \pm 0.15$ | $0.48 \pm 0.2$ |
| Maze | $0.78 \pm 0.21$ | $0.87 \pm 0.05$ | $0.57 \pm 0.18$ | $0.15 \pm 0.08$ |
| Maze2 | $0.5 \pm 0.24$ | $0.67 \pm 0.18$ | $0.65 \pm 0.15$ | $0.23 \pm 0.15$ |
| Maze3 | $0.79 \pm 0.14$ | $0.9 \pm 0.08$ | $0.95 \pm 0.05$ | $0.56 \pm 0.17$ |
| **Mean** | $0.79 \pm 0.04$ | $0.76 \pm 0.04$ | $0.74 \pm 0.04$ | $0.58 \pm 0.05$ |

Note however that all of these ablations still outperform the next best baseline (PLR, mean = 0.69). Finally, the ablation using additional DR levels instead of edited ones performs poorly, showing the strong performance for ACCEL comes from editing rather than the structure of the algorithm.

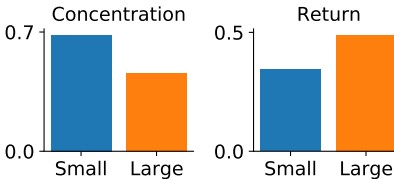

Figure 21: Replay buffer diversity vs. return in the lava environment. On the left we show the concentration of the replay buffer, measured as the percentage of the top 100 high-regret levels that can be produced by just ten parents. On the right we compare the average return on ten-tile test environments. "Small" corresponds to a buffer of size 4k, with no generator, while "Large" indicates a buffer of size 10k, using a generator 10% of the time.

**Diversity of the Level Buffer** we compare a buffer of size 4k with no DR sampling against our method with a buffer size of 10k and 10% sampling. The plots show the proportion of the top 200 levels produced by just ten initial generator levels, with a significant increase for the smaller buffer. We also compare the performance of the two agents on test levels with ten tiles, showing clear outperformance for the lower concentration agent. It is likely that hyperparameters alone will not be sufficient if we want to scale ACCEL to more open-ended problems, which we leave to future work.

## B  IMPLEMENTATION DETAILS

In this section we detail the training procedure for all our experiments. Training for all methods is conducted on a single V100 GPU, using ten CPUs. The codebase is built on top of open source repos. For PAIRED and Minimax we use results from the authors of Jiang et al. (2021a).

### B.1  ENVIRONMENT DETAILS

**Learning with Lava** The MiniHack environment is an open-source Gym (Brockman et al., 2016) environment which provides a wrapper around the full game of NetHack via the NetHack Learning Environment Küttler et al. (2020). MiniHack allows users (or agents) the ability the fully specify environments leveraging the full game engine from NetHack. For our experiments we use a simple 7x7 grid, and allow the agent to place lava tiles in any location of their choice. We considered three different DR parameterizations in Figure 19. These are: "Fixed" which always places ten lava blocks, "Binomial" which places twenty blocks, with a 50-50 chance of the block being lava or nothing, and "Uniform" which selects the number of lava blocks to place from the range $[0, 20]$. The reward is sparse, with the agent receiving +1 for reaching the goal, with a per timestep penalty of 0.01.

**Partially-Observable Navigation** Each maze consists of a $15 \times 15$ grid, where each cell can contain a wall, the goal, the agent, or navigable space. The student agent receives a reward of $1 - T/T_{\max}$ upon reaching the goal, where $T$ is the episode length and $T_{\max}$ is the maximum episode length (set to 250). Otherwise, the agent receives a reward of 0 if it fails to reach the goal.

Table 5: Total number of environment interactions for 20k PPO updates.

| PPO Updates | PLR | ACCEL (DR) | ACCEL (Empty) |
|---|---|---|---|
| 20k | 327M | 347M | 369M |

### B.2 Environment Design Procedure

The environment design procedure works as follows: at each timestep the adversary agent receives an observation consisting of a map of the entire level and then takes a two dimensional action, consisting of an object and a location, which can be anywhere in the grid. This is similar to several recent works Dennis et al. (2020); Jiang et al. (2021b;a); Khalifa et al. (2020). For MiniGrid the object is always a wall. For both methods, the goal and agent location are placed in the final two steps.

When editing, the editor has five steps to alter the environment. For the lava environment we only edit to add or remove lava tiles, while for MiniGrid we allow the editor to also change the goal location. If lava or walls (or lava) are placed in the current location of the goal or agent, then these replace the goal or agent, which must then be relocated in the final two steps of the editing episode. If the editor attempts to remove the goal or agent location in the final two steps then the action is void.

### B.3 Hyperparameters

The majority of our hyperparameters are inherited from previous works such as Dennis et al. (2020); Jiang et al. (2021b;a), with a few small changes. The first, for MiniHack we use the agent model from the NetHack paper (Küttler et al., 2020), using the `glyphs` and `blstats` as observations. The agent has both a global and a locally cropped view (produced using the coordinates in the `blstats`).

For MiniHack we conduct a grid search across the level replay buffer size $\{4000, 10000\}$ for both PLR and ACCEL, and for ACCEL we sweep across the edit method from {random, positive value loss} where positive value loss equates to a learned editor. For MiniGrid we maintain the replay buffer size from Jiang et al. (2021a) and only conduct the ACCEL grid search over the edit objective, again running both {random, positive value loss}, as well as the levels to edit from {easy, batch} and replay rate from $\{0.8, 0.9\}$. For MiniGrid, we follow the protocol from Jiang et al. (2021a) and select the best hyperparameters using the validation levels {16Rooms, Labyrinth, Maze}. The final hyperparameters chosen are shown in Table 6.

Table 6: Hyperparameters used for training each method in the maze and car racing environments.

| Parameter | MiniHack (Lava) | MiniGrid |
|---|---|---|
| *PPO* | | |
| $\gamma$ | 0.995 | 0.995 |
| $\lambda_{\text{GAE}}$ | 0.95 | 0.95 |
| PPO rollout length | 256 | 256 |
| PPO epochs | 5 | 5 |
| PPO minibatches per epoch | 1 | 1 |
| PPO clip range | 0.2 | 0.2 |
| PPO number of workers | 32 | 32 |
| Adam learning rate | 1e-4 | 1e-4 |
| Adam $\epsilon$ | 1e-5 | 1e-5 |
| PPO max gradient norm | 0.5 | 0.5 |
| PPO value clipping | yes | yes |
| return normalization | no | no |
| value loss coefficient | 0.5 | 0.5 |
| student entropy coefficient | 0.0 | 0.0 |
| generator entropy coefficient | 0.0 | 0.0 |
| | | |
| *ACCEL* | | |
| Edit rate, $q$ | 1.0 | 1.0 |
| Replay rate, $p$ | 0.9 | DR: 0.9, Empty: 0.8 |
| Buffer size, $K$ | 10000 | 4000 |
| Scoring function | positive value loss | positive value loss |
| Edit method | positive value loss | random |
| Levels edited | batch | easy |
| Prioritization | rank | rank |
| Temperature, $\beta$ | 0.3 | 0.3 |
| Staleness coefficient, $\rho$ | 0.5 | 0.5 |
| | | |
| *PLR* | | |
| Scoring function | positive value loss | positive value loss |
| Replay rate, $p$ | 0.5 | 0.5 |
| Buffer size, $K$ | 10000 | 4000 |

