# OpenReview forum: "That Escalated Quickly: Compounding Complexity by Editing Levels at the Frontier of Agent Capabilities"
_ICLR.cc/2022/Conference — ICLR 2022 Submitted_

### Official Review · Reviewer_Sh63 · 2021-10-26

**Correctness:** 4
**Technical Novelty And Significance:** 2
**Empirical Novelty And Significance:** 3
**Recommendation:** 5
**Confidence:** 3

**Main Review:**

## Strength
I am convinced that this is generally a useful problem to solve, and the existing results seem good.
The paper is easy to follow, the method is clearly explained, and the experiment is well presented.

## Weakness

I would say the environment considered are too simple, where the editing functions are fairly restrictive to the kind of "changing the landscape of the level" kind. I would have liked experiments where the dynamics of the game can be altered, for instance, the game of pong or break out with additional balls in play, or maybe mario or minecraft where the agent and the world interacts in richer ways. If we are calling them levels, I truly expect it should work for a more extensive level editor.

It is still unclear to me why the proposed method solves the problem of not having the editor generating levels that are straight up unsolvable. Or, put it in another way, how do you measure regret of a newly generated level without access to pi* ? Maybe I mis-read part of the paper but I could not find the answer to this fairly important question. Perhaps the answer to this question should be highlighted further?

minor:
I appreciated the consistent usage of pink to indicate ACCEL results, making the result section easy to read.

**Summary Of The Paper:**

A simple method of curriculum design by editing existing levels solvable by the agent to more challenging ones. Empirically it performs well, as it resolves the difficulty of randomly searching globally for a challenging curriculum by building on already challenging ones incrementally.

**Summary Of The Review:**

there is not much to say except that this is an intuitive solution to a good problem. the results over existing tasks look promising, but it would really benefit from more complex environments and edits, where the agents and environments interact in more ways than simply pathing.

---

> ### Author Response · Authors · 2021-11-19
> **Thank you for your review!**
>
> We thank the reviewer for their time and for providing us with constructive feedback. We greatly appreciate the reviewer commenting that we propose “an intuitive solution to a good problem”, and we hope that in light of this we are able to receive a higher score once we have clarified the concerns.
>
> > Environment too simple
>
> We respectfully disagree that this environment is too simple:
> It has 147d observations with a 52d environment design space. Our agents train from pixel-like observations with a ResNet+LSTM policy for >300M steps.
> It was used as the primary experiment in both PAIRED and PLR, two prominent UED algorithms that we use as our baselines.
> It has been used in several state of the art exploration papers as their primary benchmark, for example RIDE, AMIGo, BeBold, AGAC, all of which have been published at top tier venues in the past two years.
>
> We believe that *not* using this environment would be highly questionable given its prominence in the literature. However, we do acknowledge that in general it is important for UED to extend to new environments.  We agree looking into further environments for UED is interesting, but outside of the scope of this paper. Note that level editing works in the games community typically focus on grid worlds (e.g. Khalifa et al 2020). Extending to environments like Minecraft would be interesting, but would likely require additional tools which are not yet available. We think extending to this type of environment is exciting future work.
>
> Furthermore, we also have a second environment which is entirely different, consisting of a global observation and lava tiles which cause instant death, posing a challenge for environment design methods.
>
> > Editor producing solvable levels
>
> Note that we are using *approximate* regret. These approximations were introduced in Robust PLR (Jiang et al 2021). In this paper we use “positive value loss” which is equivalent to the sum of positive advantages from a trajectory. Thus we can approximate regret with rollouts from the current policy.
>
> Equipped with this approach for scoring levels with approximate regret, the student evaluates *but does not train* on levels produced by *both the editor and the generator*. Thus, if a level is unsolvable, it will have low (likely zero) regret and thus will not be selected by the Curator. This means the student only trains on solvable levels, which also means we inherit the theoretical results from Robust PLR, that at equilibrium the student is following a minimax regret strategy.
>
> This *should* be explained in Section 2.2 and 3. Please let us know if we need to make additional clarifications in the paper.
>
> Finally, we are glad our plots were easy to read!

---

> > ### Comment · Reviewer_Sh63 · 2021-11-22
> > **thanks for the response**
> >
> > I think having this exact phrase "Thus we can approximate regret with rollouts from the current policy." would probably settle a lot of the confusion that I had.
> >
> > Overall after reading the other reviews, it does seem that the majority of the complaints are the scope of this approach, i.e. how do we scale this to a more challenging domain. I am unsure how this can be addressed, but given the proposed method is "simple", then it seems it will be of value to prove its worth by running it under a different domain and with more computing power.
> >
> > From an author's POV, I definitely can empathize as I too find this aspect of the review process (of demanding a more challenging benchmark / tasks) frustrating. I feel a good idea should merit a good reception from the reviewers even if the experiments are not as thorough as they have demanded. However, often times there is no good way to check if an idea is good unless it can be communicated through a more thorough set of experiments.

---

> > > ### Author Response · Authors · 2021-11-23
> > > **Thank you for the suggestion!**
> > >
> > > Thank you for this suggestion. You may have seen already but we just updated our manuscript. Since it is now very close to the deadline, we cannot do this again, but we will most certainly include the sentence proposed in the next version of our paper.
> > >
> > > We strongly disagree that our experiments are not *thorough*. We ran multiple competitive baselines, and ran/reported a series of ablation studies. In the Appendix we show qualitative and quantitative analysis of our method, explaining how and why it works. Our zero-shot results include 12 tasks, and we used robust statistics such as IQM, where our method shows significant gains over all baselines. Further, we tested the limits of the approaches with a 51x51 and a 101x101 grid which is even challenging for humans since it is partially observable. Several baselines completely fail on the 51x51 PCG maze, demonstrating that it is non-trivial. The only method that does not fail is Robust PLR with 25% success rate, compared to ACCEL with 53%. If you are able to view the video on our [website](https://sites.google.com/view/compoundingcomplexity), ACCEL learns very general behavior. Thus, **we believe we have been thorough with our experiments**.
> > >
> > > We also contest the idea that the other environments proposed by reviewers constitute "scaling to a more challenging domain". We certainly agree that continuous control is sufficiently *different*, that it would show our method is more broadly applicable. However, it is subjective (and understudied) to quantify what is more challenging: a partially observable navigation task with a sparse reward, learning from an ego centric view vs. a dense reward MuJoCo environment from a proprioceptive state. It seems the reviewers tend to believe the latter is more challenging, which has been our major pushback. All we can do here is point to the list of empirical works that have used MiniGrid environment in the community, which is visible on the README of the [GitHub repo](https://github.com/maximecb/gym-minigrid). Note that our agent learns to solve all of these tasks without access to any intrinsic reward, which is typically used extensively in MiniGrid.

---

### Official Review · Reviewer_MHvz · 2021-11-02

**Correctness:** 2
**Technical Novelty And Significance:** 2
**Empirical Novelty And Significance:** 2
**Recommendation:** 5
**Confidence:** 4

**Main Review:**

Strengths:

The main idea of the paper is intuitively reasonable and the experimental section shows promising performance on the investigated tasks. The related work section is extensive and adequately situates the conducted research in a broad context.

Weaknesses:

Although the actual idea behind the paper is very incremental, the reader is still left with questions particularly w.r.t. details of the method and Remark 1:
- In Remark 1, the authors state that if their method finds a Nash Euqilibrium, then the student policy is following a minimax regret strategy. I understand that the authors take this result from the Robust PLR paper, to which their method is very similar. However, I have certain concerns regarding this statement:
    - The proofs in the Robust PLR paper are relying on the assumption that the replay teacher is utilizing the true regret to guide its sampling. However, there is no guarantee that either the MaxMC or PositiveValueLoss are actually estimating the true regret in a realistic setting. If e.g. an agent is not observing any reward signal in a sparse reward task, it can have zero residuals by predicting a value of 0 in all states, although there may exist a policy solving the task. In this case, MaxMC and PositiveValueLoss would not estimate the true regret.
     - By using the level editing scheme, the second teacher in the dual curriculum seems to employ a different utility function than the constant one assumed in Robust-PLR. Does this change the theoretical results?
     - These disconnects between the theory in the Robust-PLR method and the presented method should be carefully addressed in order to avoid false conclusions from reader of the papers.
- Looking at Table 5, it is unclear whether the additional environment interaction required to estimate the regret on the edited levels is taken into account in the presented results. Table 5 already shows that the number of environment steps differs by around 10% between methods. Can the authors explain the reason for this difference? Further, the plots in the main paper use "Student PPO Updates" as the x-axis unit. I guess that this unit only takes into account the environment steps taken in levels WITHOUT the stop gradient, i.e. does not take into account the additional required evaluations incured by the level generator? Given that multiple new levels can be generated for each level selected for student training, this could be a high hidden cost.
- From the statements at the beginning and end of Section 4.2, it seems that ACCEL does not use the random level generation, as the authors state that "for ACCEL we begin with empty rooms and randomly edit the block locations" and further investigate an ablation where they edit uniformly sampled levels. If the regular ACCEL algorithm is not using the domain randomization, Algorithm 1 yields a wrong picture of the ACCEL algorithm, as the random sampling case will never take place in practice.

Given the large amount of space that the authors spend on high-level introduction and discussion w.r.t. related work, it seems unsatisfying that these technical details are not clarified in the main paper.

Minor Points:
 - Question: Would it be possible to compare to some adaptive domain randomization methods that do not just randomly sample tasks? Or are they ill-suited for the investigated experiments? If so, why? It would be interesting to see how a method following a different methodology performs in these experiments.
 - Another related work that should be mentioned is the work on contextual MDPs [1], which is conceptually similar to the UPOMDPs used to model the space of possible tasks.

[1] Modi, Aditya, et al. "Markov decision processes with continuous side information." Algorithmic Learning Theory. PMLR, 2018.

**Summary Of The Paper:**

The work proposes an extension to the recently proposed "Robust Prioritized Level Replay" (R-PLR) method that aims to train RL agents capable of generalizing to a large range of parameterized tasks by curating a curriculum over training tasks. The authors show that by editing levels that are at the frontier of the current agent capabilities instead of proposing novel ones by random sampling, the performance of agents trained with R-PLR can be further increased.

**Summary Of The Review:**

The incremental nature of the method in combination with the unclarity w.r.t. details and theory behind the method are not allowing me to recommend acceptance. I may improve my score if my concerns are adequately addressed.

---

> ### Author Response · Authors · 2021-11-19
> **Thank you for your review!**
>
>
> We thank the reviewer for commenting that our method is intuitive and for appreciating our attempt to provide a thorough literature review. It seems the majority of the reviewer's concerns are around the differences in theoretical results when using the editor. We hope we can sufficiently clarify these concerns below.
>
> > no guarantee that either the MaxMC or PositiveValueLoss are actually estimating the true regret in a realistic setting
>
> We agree with the reviewer that these two approaches will not guarantee a perfect estimate of the true regret, which is often computationally intractable. Extending the analogy provided with a sparse reward environment, these two approximate regret estimates will only be greater than zero if the agent is able to find the sparse reward. Which means the curator will select environments where the agent is initially close to the goal, without obstacles. This is a good thing, as it means the agent quickly learns to navigate towards the goal, likely much faster than if it trained on the full distribution, containing many environments that do not provide any signal for learning.
>
> > the second teacher in the dual curriculum seems to employ a different utility function than the constant one assumed in Robust-PLR. Does this change the theoretical results?
>
> Since the student agent *does not train on levels produced by the editor*, and only trains on levels selected by the curator, there is no difference between the theoretical result for ACCEL and Robust PLR.
>
> > PPO Updates in the x-axis
>
> The reviewer is correct that when we use the stop gradient this is not counted towards a PPO update so the plots show performance with differing numbers of *environment interactions*. We felt that it made the most sense to compare the curriculum learning methods in terms of the quality of the curriculum - ie - *how much does the student learn from the same number of gradient steps*, where the only difference is the training level.
>
> If this is still unsatisfactory, note that ACCEL only uses ~10% more data than Robust PLR, yet significantly outperforms it. Furthermore, in our ablation study we used the 10k Update checkpoint (so that we could run multiple ablations), and ACCEL even at 10k significantly beats Robust PLR (0.79 vs. 0.71). The gains at 20k updates are significantly larger.
>
> > Correctness of Algorithm 1
>
> This is a good question. The algorithm is still correct as **the empty rooms are randomized**, wrt the agent and goal locations. It is essentially DR with zero blocks (which could also be sampled by our DR and PLR baselines, since they select between 0-60 blocks). To correct this, we will include the generator distribution as an input to the algorithm. We hope this is clear now, if not then please let us know!
>
> > Active Domain Randomization methods.
>
> We did not intentionally gloss over these methods, and will be sure to cite them in the next version of our paper. However, since we used the environment from PAIRED and Robust PLR, we also used the baselines, which have grown as new methods have been added. We compare against both of these methods, plus all of the baselines they each considered. We feel this is thorough and cannot consider all possible baselines.
>
> > Contextual MDPs
>
> Great point, we will make the necessary references here.

---

> > ### Comment · Reviewer_MHvz · 2021-11-26
> > **Thank you for the clarifications**
> >
> > I thank the authors for the clarifications. My main concern is still w.r.t. the statement made about the optimality of the algorithm in Remark 1.
> >
> > > We agree with the reviewer that these two approaches will not guarantee a perfect estimate of the true regret, which is often computationally intractable.
> >
> > I agree with the explanations of the authors. However, I believe that Remark 1 and the sentences on the beginning of page 5 are too strong and should hence be removed or explained in much greater detail. Saying that "ACCEL uses a minimax regret objective" is problematic as readers not familiar with the details of the algorithm may just assume that the MaxMC and PositiveValueLoss are correct estimators of agent regret. In combination with Remark 1, those readers may then assume that ACCEL is a full solution to the minimax regret problem, which, as the authors say in their answer, it is not because regret is intractable to estimate.
> >
> > > The reviewer is correct that when we use the stop gradient this is not counted towards a PPO update so the plots show performance with differing numbers of environment interactions. We felt that it made the most sense to compare the curriculum learning methods in terms of the quality of the curriculum - ie - how much does the student learn from the same number of gradient steps, where the only difference is the training level.
> >
> > I see. And I agree that this choice of x-axis is a well-motivated in the context of the paper. Maybe it is possible to make this difference in environment interactions more obvious via a footnote that points to an appendix for in-depth discussion.

---

### Official Review · Reviewer_Xz9V · 2021-11-02

**Correctness:** 4
**Technical Novelty And Significance:** 2
**Empirical Novelty And Significance:** 3
**Recommendation:** 5
**Confidence:** 3

**Main Review:**

This paper puts forth compelling evidence that it is the state-of-the-art UED algorithm, at least within experimental domains it considers. Its suite of baselines (DR, PAIRED, PLR, Minimax adversarial) are to my knowledge well-chosen: PAIRED and PLR represent prior state-of-the-art UED approaches, and DR and Minimax adversarial simpler baselines. It highlights, by certain measures, higher levels of emergent complexity as well as superior performance on heldout environments both within the training distribution and outside of it. While some comparisons (esp. the Lava ones) show modest gains, others show more meaningful ones, especially the PerfectMaze heldout environment performance. Aside from the main comparison, it contains ablation studies helpful for understanding the utility of various choices made.

The paper is well-written and fairly easy to understand. The authors make effective summaries: Figure 2 and Algorithm 1 (especially with the highlighted difference from PLR, which can be confusing otherwise) are useful, and Table 1 is helpful as well.

My main criticism is that the evaluations seem somewhat limited in scope. While Lava and MiniGrid comparisons of this form are standard for UED methods, they are not the only ones, and in particular, they seem to be the most obvious contenders for an editing-type approach to be successful. As the paper points out, these environments have a natural empty-room starting point from which to build the curriculum and edit off of. These edits naturally seem to curricularize and, small edits easily make for what can be readily interpreted as reasonable additions for training: edits that functionally change how an agent must explore, encouraging exploration generally, or data augmentations that change irrelevant bits. Other benchmarks that existing works use (CarRacing, MuJoCo Hopper, Procgen) intuitively have a different flavor. This makes me wonder whether this method is useful beyond these sorts of environments that this sort of editing process is intuitively suited for.

The ablation study also raises questions. If I am interpreting this correctly, it seems to show that, at least on MiniGrid metrics, several versions with pieces stripped out perform statistically very similarly, except when editing is removed, which substantially drops performance. This makes me think two things. (1) It heightens my worry about the limited scope of evaluations, where editing seems tailor-made to succeed. (2) It makes me wonder whether something a good deal simpler than that proposed algorithm works just as well.

One choice that I found odd was the inclusion of Figure 4 in the main text but the placement of Figure 15 in the appendix. While emergent complexity is interesting to see, it is hard to judge whether obtaining higher scores on these metrics is objectively "good" (and e.g. I can imagine ways in which adding more lava tiles is not particularly useful towards an end performance objective. Figure 15 is, on the other hand, a more straightforward comparison of methods.

Minor: Figure 4 color choices are confusing. If I am reading correctly, (b) contains information entirely about the ACCEL runs, but it uses the same color scheme as the baseline.

**Summary Of The Paper:**

This paper introduces Adversarially Compounding Complexity by Editing Levels (ACCEL). ACCEL is an Unsupervised Environment Design (UED) algorithm, a method of generating a curriculum of environments so as to train agents that generalize well to either a training distribution of environments or off-distribution environments. ACCEL bares similarity to a recent addition in the UED literature, Robust PLR, but importantly uses an editor to modify previously seen environments. Edited levels are used if they satisfy a criterion based on the PLR score, and hence this can be thought of as a sort of evolutionary algorithm.

After introducing ACCEL, the paper presents a series of experiments in the Lava grid and Minigrid environments, benchmarking ACCEL against a suite of UED and simpler baselines. It demonstrates emergent complexity generated in algorithms' curricula (with metrics such as number of lava tiles/blocks, shortest path length) as well as generalization to held-out levels.

**Summary Of The Review:**

This paper presents a novel UED approach that presents performance gains on several baselines. My chief concern is that the benchmarks are limited in scope to situations that seem most likely for an editing process to succeed, whereas prior work demonstrates efficacy in domains that seem less suited for editing approaches. I recognize that this is an intuitive judgement about the benchmarks on my part; if the authors can put forth a compelling argument that the experiments are not so limited in scope, I would be inclined to raise my score.

---

> ### Author Response · Authors · 2021-11-19
> **Thank you for your review!**
>
> We thank the reviewer for their overall positive review. It seems the reviewer’s concerns are very concentrated on our choice of environments. We will attempt to address those concerns below, and hope that this is sufficient to see our score increased.
>
> > Environment favoring editing.
>
> We would like to point out that editing is not a specific method at all. Indeed, *editing is an evolutionary approach, arguably one of the most general approaches known.* All of the environments mentioned by the reviewer could be represented in this way - CarRacing could begin with a round track and edit to add corners, and MuJoCo environments could begin with the regular environment and edit to add torques/alter physics or morphology.
>
> We prioritized the MiniGrid experiment since it has been used in both PAIRED and PLR, recent state of the art UED methods. We believe it would be highly questionable if we had *not* used this environment to benchmark our new UED algorithm building on ideas from those works. ACCEL produces large gains in this environment, and our analysis clearly shows the benefit comes from the combination of editing and regret-based curation, which consistently extends the frontier. As a second, smaller experiment we included a different gridworld environment using lava and fully observable observations, and prioritized ablations of our method to fully demonstrate its robustness, rather than adding a third environment.
>
> Finally, we note that grid-like games such as these are used extensively by level generating works for example many of those from Togelius et al. as well as used almost exclusively for exploration methods published at top venues (such as ICLR), e.g. BeBold, RIDE, AGAC, AMIGo.
>
> > Questions from the ablation study
>
> The reviewer is correct that the ablation study shows the editing method is somewhat agnostic to the more arbitrary design choices we made. We were very reassured by this ablation and certainly do not see it as a negative. Conversely, **we believe this reinforces the generality of the method**. However, what is novel about our approach is introducing a new hybrid algorithm that *uses regret as a “fitness” function* and thus provides theoretical results as well as strong empirical performance. We are surprised to hear of the reviewer comment that something simpler may work better, since in our view (and from other reviewers) our method is simple - especially compared with POET which requires an entire population of agent/environment pairs, and PAIRED which requires a learned adversary and an additional antagonist agent to get similar regret guarantees at equilibrium. ACCEL only trains a single agent.
>
> > Figure 4. (and Figure 15)
>
> We understand the reviewer’s concern here. We made the choice because we wanted to emphasize in the first experiment that in some settings editing can facilitate learning in challenging design spaces, where many of the randomly sampled levels are too hard for an early stage agent. This could have implications for safe learning, for example in design spaces containing a high percentage of levels where the agent “dies”. This is directly seeking to answer the first Q. It is also worth noting that we do indeed outperform in Figure 15, so we were not hiding anything aside from a set of strong results. We believe that we included a sufficiently wide variety of zero-shot transfer results in 4.2, which is the real focus of our empirical results as they are much larger experiments.
>
> > Figure color choices
>
> We agree, the bars do not need to be different colors. We have changed this in the updated manuscript, thanks!

---

> > ### Comment · Reviewer_Xz9V · 2021-11-23
> > **Thanks for the response!**
> >
> > Thank you for your careful response.
> >
> > Regarding choice of environments:
> >
> > -I agree that it was important to include the environments that you did include, and that the evidence shown there is quite informative.
> >
> > -I also agree that editing is in principle a very general approach, and that one could conceivably extend the approach successfully to the other benchmarks mentioned. I think it would be much more convincing if you actually do demonstrate this. Precisely how one introduces edits in these other benchmarks seems like the critical question. In the environments you used, how one does this seems straightforward relative to what one would need to do in the environments you did not use.
> >
> > -I also agree that many papers in top venues such as this exclusively use environments such as that which you used. However, your chief comparisons include extensions to other environments, so it would be much more convincing if you did, too.
> >
> > I'll echo Reviewer Sh63 that it can be frustrating to get the feedback that one's experiments should include more benchmarks -- there are always more benchmarks to run. However, in this case, I think the additional benchmark request is for a principled reason. Hence, I am not inclined to change my score.
> >
> > A more minor point, I'll agree that the ablation study can be taken to reinforce the generality of the method. What was not clear to me, on a first read, was whether it suggests that an even _simpler_ method than that proposed in the main text could yield comparable performance. I agree that your method is quite simple, and that that is a real strength! It might just be helpful to have a short statement clarifying how the ablation still shows that each component of the proposed method (or some reasonable variation of each component) is required for the whole thing to work.

---

### Official Review · Reviewer_dd8g · 2021-11-04

**Correctness:** 3
**Technical Novelty And Significance:** 2
**Empirical Novelty And Significance:** Not applicable
**Recommendation:** 6
**Confidence:** 4

**Main Review:**



**Strenghts:**

- The idea is straight-forward.
- The paper is well-written.
- Table 1 is a nice comparison
- S.4 The results seems mostly good. (On some levels, DR performs on-par and that's never really explained)

**Weaknesses:**

- W.1 I think it's hard to generalize this method to other domains without a lot of domain knowledge on what constitutes small incremental changes for the agent. For example I could see this being applied to generate a curriculum for a robot arm to learn grasping different objects by incrementally deforming a mesh but I think a lot of trial-and-error would be required to find the right deformation size that's enough change to be interesting but not enough to make grasping the object physically impossible. So maybe the authors could give a few examples of how this applies to other possibly non-discrete domains or if the paper should get rejected, include an experiment in a non-gridworld setting (i.e. something where it's not trivial what the incremental changes are), like the POET paper.
- W.2 Clarity - (a) Is Fig.4 reported on a held-out test set or during training? Because if it's from during training, the level difficulty would dictate the meaning of the return plot. So maybe a different metric would be better like return divided by  difficulty metric (like shortest path length) or something like that. (b) How do you calibrate the threshold that the PLR score has to match?  (c) In Fig.6, why is DR performing on-par with ACCEL in 3 cases?

**Nitpicks & Questions:**

- Q.1 Fig.2 is well-made but unintuitive and I don't think in its current state it does a great job at summarizing your method: (a) The order of things is unclear. Adding numbers to indicate the order of operations would be helpful. (b) The generator isn't really crucial in that sure, it creates the initial set of levels but shouldn't it generate them into the level buffer on the left? From the current diagram it's not clear if the generator is a one-off or a recurring thing. (c) What's the "Curator" in practice? The level buffer + threshold? I.e. if a given level has a PLR value above the threshold, it's considered ready for play? This is a very personal opinion but I think "Curator" implies more of an active system than what's currently there. (d) Why is there a red-tinged level in the level buffer? (e) Why are there 2 arrows going from the student into the level buffer? (f) What is "update buffer" from the student?
- Q.2 I feel like you're glossing over a lot of non-uniform domain randomization literature that's also maintaining a population of environments and tries to make them harder one step at a time like ADR, and SS-ADR [1,2]. Wouldn't these methods also work (similarly) to generate the curriculum for your agent?

**References:**

- [1] Mehta, Bhairav, Manfred Diaz, Florian Golemo, Christopher J. Pal, and Liam Paull. "Active domain randomization." In Conference on Robot Learning, pp. 1162-1176. PMLR, 2020.
- [2] Raparthy, Sharath Chandra, Bhairav Mehta, Florian Golemo, and Liam Paull. "Generating Automatic Curricula via Self-Supervised Active Domain Randomization." arXiv preprint arXiv:2002.07911 (2020).



**Summary Of The Paper:**

The paper introduces the idea of adding small incremental changes to grid world levels, to create an automatic curriculum for navigating agents.

**Summary Of The Review:**

The paper feel like it's a small incremental change to existing literature and since I can't really imagine how this is going to generalize to non-gridworld domains, I can't recommend acceptance at the moment. If the authors could maybe include additional discussion of how their method applies to other environments or better, include experiments in non-gridworld settings, I would change my rating.

---

> ### Author Response · Authors · 2021-11-19
> **Thank you for your review!**
>
> We thank the reviewer for the constructive feedback. We are pleased to see the reviewer appreciated the idea is simple and our paper is well-written. We hope to address your concerns below, and kindly ask that you consider raising your score if our responses are satisfactory.
>
> > W.1 I think it's hard to generalize this method to other domains without a lot of domain knowledge on what constitutes small incremental changes for the agent
>
> We agree that it is important to specify the parameterization for the environment, but we respectfully disagree that this is unique to editing. For instance, DR and other methods will also fail if their environment parameterization is not effective. See the Appendix (Fig. 19a) and b)) where we explored the impact of DR parameterization on the performance of a DR agent. Indeed, since our method is an evolutionary algorithm it is very general and could be applied in many domains.
>
> This combination of evolution and a regret-based curator allows us to produce batches of new environments that are incrementally harder/easier than a base environment, **while only training the agent on those at the frontier of its capabilities**. Thus, the method is not overly sensitive to the parameterization - as long as some levels are useful, we still continue to expand the frontier. This is why all ablations also work effectively.
>
> The reviewer mentioned manipulation, but ACCEL could also be used to alter terrain/physical properties or morphology in a locomotion environment.  We did not conduct these experiments because we instead focused on MiniGrid, which is a prominent environment used in UED (and elsewhere). Note that MiniGrid is not a toy grid world: the observation space is >100 dimensions compared to <20 for locomotion tasks from proprioceptive states. We use a ResNet+LSTM policy and train for >300M steps, which is a larger architecture and training time than typically used in locomotion environments. Furthermore, the results we get beat the previous two algorithms (PLR and PAIRED) presented at NeurIPS 2020 and NeurIPS 2021, using this same environment, providing consistent gains over recent works.
>
> > W.2 Clarity
>
> Thank you for highlighting areas where we need to improve our clarity. See responses below:
>
> a) This is during training, and we agree, but note that ACCEL has the highest return *and* most difficult environments. We felt this was clear to see. For the test performance see the Appendix A.3. **ACCEL is best in every test environment.** We believe the training dynamics were most interesting for this experiment, since it may have implications for training agents in safety critical settings.
>
> b) The PLR score threshold is dynamic, compared to the existing buffer. E.g. if the highest N levels in the buffer have regret from high to low $[x, …, x-\epsilon]$ then any new level with regret above $x-\epsilon$ will be accepted in the buffer, kicking out the bottom entry.
>
> c) This zero-shot benchmark includes every environment proposed by prior works, representing a broad capability of agents in MiniGrid. Some are simpler than others, for example SmallCorridor is solved by DR, PLR and ACCEL.  We respectfully disagree with the idea that baselines should not perform well on **any** of the zero shot environments. Indeed, in many papers this is the case, for example in PAIRED (Dennis 2020) the DR agent performs as well as PAIRED in 3/6 environments, *but crucially not on the harder tasks*. We believe the fact ACCEL outperforms on 9/12, and has much stronger aggregate performance (Fig. 7b)) is very convincing.
>
> > Q1. Figure 2
>
> Thank you for commenting that the Figure is well made and providing constructive comments to help improve it.
>
> a) We can certainly add some numbers.
>
> b) The generator does not go directly into the level buffer, we only choose the generated levels with high regret. This can be meaningful, for example if the agent and goal are close together then the task is much simpler. Even empty rooms can be challenging for the agent initially, since it is partially observable.
>
> c) The reviewer is correct that the Curator is the level buffer + threshold, but note that it is an active strategy: The Curator samples from the buffer using a bandit-like approach, since the scores are non-stationary. This is discussed in detail in PLR (Jiang et al 2021).
>
> d) The red and green colors in the level buffer indicate where they came from. The red ones are directly from the generator and not edited.
>
> e) Arrows from the student indicate the student’s behavioral metrics collected on each level are used as a prioritization signal for whether each level should be added to the PLR buffer.
>
> f) After the student is evaluated on a level the score updates.

---

> > ### Author Response · Authors · 2021-11-19
> > **2/2**
> >
> > > Q2. Active Domain Randomization methods.
> >
> > We did not intentionally gloss over these methods, and will be sure to cite them in the next version of our paper. However, since we used the environment from PAIRED and Robust PLR, we also used the baselines, which have grown as new methods have been added. We compare against both of these methods, plus all of the baselines they each considered. We feel this is thorough and cannot consider all possible baselines.
> >
> > Thank you for your time!

---

> > ### Comment · Reviewer_dd8g · 2021-11-22
> > **Comment on Rebuttal**
> >
> > Dear Authors,
> >
> > Here is some feedback on your rebuttal:
> >
> > ---
> >
> > > MiniGrid, which is a prominent environment used in UED
> >
> > Sure, but both PAIRED and Robust PLR also show that their method generalizes to other environments (Hopper and CarRacing respectively). And POET is shown in an entirely different environment (2D Walker with continuous action) which is a harder control problem in my humble opinion.
> > So I'm asking if you're so certain that your method generalizes, why didn't you include any other environments?
> >
> > ---
> >
> > >  train for >300M steps
> >
> > and
> >
> > > We use a ResNet+LSTM policy
> >
> > That sounds like it's a bit much. For example, OpenAI in their Hide&Seek paper [1] got really complex behavior in 3D continuous control agents happening after 300 mil steps. Do you think that's a function of the complexity of the problem or a function of the number of parameters in your network? I have a suspicion it's the latter, which is why DRL networks are traditionally very shallow and from my personal experience, using a ResNet was detrimental to performance. Have you tried e.g. an EfficientNet as a feature extractor with frozen weights and only training the RNN? The other question I now have is if the Resnet is even necessary - meaning your agent is not looking at pictures of cats and dogs but a fixed grid. There can only be one of 4-5 values in any given pixel, right? And so maybe an architecture that's closer to CIFAR-10 would perform better in terms of sample efficiency and allow you to run other experiments too.
> >
> > ---
> >
> > > the observation space is >100 dimensions compared to <20 for locomotion tasks
> >
> > I don't think you can make a case that a grid world environment is harder than a continuous control task because the observation is 2D and not 1D.
> >
> > ---
> >
> > > We believe the training dynamics were most interesting for this experiment, since it may have implications for training agents in safety critical settings.
> >
> > I don't follow. Can you explain, please?
> >
> > ---
> >
> > > Re Figure 2
> >
> > These comments weren't necessarily supposed to be answered in the rebuttal - I was just trying to illustrate what questions I have as a reader just by looking at this figure. And yes, if you could number the different paths and explain what the order of operations is, that would greatly benefit clarity.
> >
> > ---
> >
> > > Re my review score:
> >
> > I think all reviewers pointed out similar issues of clarity and scope of this solution. In its current form, the focus seems to be too narrow and despite claims of generality, you don't actually show that in the paper. I think you'd greatly benefit from additional non-gridworld environments.
> > However, as the authors point out, the method outperforms the baselines on all the gridworld tasks and that is a valid contribution. I personally don't think it's a very exciting contribution but since you're improving upon the baselines, I'm changing my rating to a borderline accept.
> >
> > ---
> >
> > One more general remark: I think reviewers usually appreciate it if you update the manuscript after writing the rebuttal to incorporate some of the feedback the reviewers gave.
> >
> > ---
> >
> > ### References:
> >
> > - [1] Baker, Bowen, Ingmar Kanitscheider, Todor Markov, Yi Wu, Glenn Powell, Bob McGrew, and Igor Mordatch. "Emergent tool use from multi-agent autocurricula." arXiv preprint arXiv:1909.07528 (2019).

---

> > > ### Author Response · Authors · 2021-11-23
> > > **Thank you for your comments!**
> > >
> > > We thank the reviewer for a quick response to our rebuttal, and for providing us with additional opportunities to clarify our work. Further, we appreciate the score raising as a result of our strong results vs. baselines. See our more detailed responses below.
> > >
> > > > Why didn’t we include any other environments?
> > >
> > > We acknowledge that our paper does not have a continuous control environment, as we focused our resources on the MiniGrid experiment. We chose to use another grid-world environment for our second experiment, where the challenge was altered due to being fully observable with potential for instant death. We did not run a third environment simply due to the amount of resources it would require. Given our existing experiments show large gains in two settings we believe this is sufficient.
> > >
> > > > Number of Training Steps
> > >
> > > We want to begin by saying that we found the results from the OpenAI paper to be very interesting. However, we disagree that there is an unfavorable comparison compared to our work. We are not aware of OpenAI achieving complex behavior at 300M steps. On the website it says that up to *episode* 2.69M, all that happens is “seekers learn to chase hiders”, the simplest behavior. In the paper it says an episode is 240 steps, so this is ~645M steps, over double what we quoted. Furthermore, the OpenAI agent trained from x,y coordinates, which is a simplified observation compared to using an agent-centric crop of the grid as an input without any inductive biases about the location of the goal/agent. As an additional point of reference, the PAIRED paper used around 3Bn steps in this same MiniGrid environment.
> > >
> > > > Policy Architecture
> > >
> > > The ResNet policy we use is standard in vision based tasks, for example OpenAI Procgen, it was originally introduced in the IMPALA paper (Espeholt 2018) and has fewer than 1M parameters. We did not try pre-training the feature extractor, but that likely sounds like a promising avenue for future work, as the controller could be simpler to learn.
> > >
> > > > Difficulty vs. Dimensionality
> > >
> > > We agree there is not a one-to-one mapping between dimensionality and difficulty, but note that many works in the field use the MiniGrid environments for exploration and UED, whereas the MuJoCo environments have largely been displaced by the pixel-based DM Control Suite. To see the list of works using MiniGrid, see the README on the GitHub repo: https://github.com/maximecb/gym-minigrid.
> > >
> > > > Safety critical
> > >
> > > If it is the case that there are obstacles in the environment which lead to death for an agent, and that is undesirable for some reason, then our method shows it is possible to first begin without these obstacles present and still quickly learn the required behavior.
> > >
> > > > Updating the Manuscript
> > >
> > > We apologize for not doing so sooner, since it required re-approval internally we are only now able to upload the new version. We made the following changes:
> > > - We switched the colors in Fig 4.b), as suggested by Reviewer Xz9V.
> > > - We now cite the ADR papers mentioned by both Reviewer dd8g and MHvz.
> > > - We now cite the Contextual MDP paper in preliminaries, suggested by Reviewer MHvz.
> > > - We included the generator as an input for the algorithm box, to clarify the issue raised by Reviewer MHvz.
> > >
> > > We hope you found these additional responses helpful.
> > > Thank you!

---

### Author Response · Authors · 2021-11-23
**Updated Manuscript**

We apologize for not updating our paper sooner, since it required re-approval internally we are only now able to upload the new version. We have made the following changes:

- We switched the colors in Fig 4.b), as suggested by Reviewer Xz9V.
- We now cite the ADR papers mentioned by both Reviewer dd8g and MHvz.
- We now cite the Contextual MDP paper in preliminaries, suggested by Reviewer MHvz.
- We included the generator as an input for the algorithm box, to clarify the issue raised by Reviewer MHvz.

Thank you all!

---

### Decision · Program_Chairs · 2022-01-20

**Decision:**

Reject

**Comment:**

This paper tackles the problem of Unsupervised Environment Design to train more robust agents. The proposed method trains RL agents by generating a curriculum of training tasks to enable agents to generalize to many tasks. The key contribution is an algorithm to generate this curriculum by incremental edits of the grid world environments. The reviewers all agreed that the paper is well-written and the method is intuitive. However, the weakness of this work is also obvious: the proposed method is only evaluated in grid worlds, and it's unclear how the editing approach can be easily generalized to more complex environments. This submission would benefit from more comprehensive evaluation in non-grid world environments, especially given that the compared baselines have results in other environments too.